# Gut microbiota impacts bone via *Bacteroides vulgatus*-valeric acid-related pathways

Xu Lin [1,2,10], Hong-Mei Xiao [3,10] ✉, Hui-Min Liu[3], Wan-Qiang Lv [3], Jonathan Greenbaum[4], Rui Gong [2], Qiang Zhang[4], Yuan-Cheng Chen[2], Cheng Peng [2], Xue-Juan Xu [2], Dao-Yan Pan [2], Zhi Chen [2], Zhang-Fang Li [2], Rou Zhou[2], Xia-Fang Wang[2], Jun-Min Lu[2], Zeng-Xin Ao[2], Yu-Qian Song[2], Yin-Hua Zhang[2], Kuan-Jui Su [4], Xiang-He Meng [4], Chang-Li Ge [5], Feng-Ye Lv[5], Zhe Luo [4], Xing-Ming Shi [6], Qi Zhao [7], Bo-Yi Guo [8], Neng-Jun Yi[8], Hui Shen [4], Christopher J. Papasian [9], Jie Shen [1,2] ✉ & Hong-Wen Deng [4] ✉

Although the gut microbiota has been reported to influence osteoporosis risk, the individual species involved, and underlying mechanisms, remain largely unknown. We performed integrative analyses in a Chinese cohort of peri-/post-menopausal women with metagenomics/targeted metabolomics/whole-genome sequencing to identify novel microbiome-related biomarkers for bone health. *Bacteroides vulgatus* was found to be negatively associated with bone mineral density (BMD), which was validated in US white people. Serum valeric acid (VA), a microbiota derived metabolite, was positively associated with BMD and causally downregulated by *B. vulgatus*. Ovariectomized mice fed *B. vulgatus* demonstrated increased bone resorption and poorer bone microstructure, while those fed VA demonstrated reduced bone resorption and better bone micro-structure. VA suppressed RELA protein production (pro-inflammatory), and enhanced IL10 mRNA expression (anti-inflammatory), leading to suppressed maturation of osteoclast-like cells and enhanced maturation of osteoblasts in vitro. The findings suggest that *B. vulgatus* and VA may represent promising targets for osteoporosis prevention/treatment.

Osteoporosis (OP), which is mainly characterized by low bone mineral density (BMD) and increased susceptibility to low trauma fractures, is most prevalent in post-menopausal women (termed postmenopausal osteoporosis (PMOP)). 4.9 million post-menopausal women in the US suffered osteoporotic fractures between 2000 and 2011, and the hospital costs associated with OP exceed those for myocardial infarction, stroke, and breast cancer[1,2]. The pathogenic mechanisms for PMOP and OP remain unclear, and current intervention and treatment options are not totally satisfactory[1,2].

Human studies have focused on the potential impacts of both inherent (e.g., (epi-)genome, transcriptome, proteome, metabolome) and external (e.g., environmental, medical/medication/nutrition,

lifestyle) risk factors for OP[3]. Recently, however, experimental animal models have identified strong associations between gut microbiota (GM) and bone. For example, Sjogren et al. found that absence of GM in mice leads to increased bone mass that normalizes following normal GM colonization[4]. In humans, several early studies have suggested the potential importance of GM for bone health[5–11]. In one of these studies, Ni et al. performed two-sample Mendelian randomization (MR) analysis with genome-wide association study (GWAS) summary statistics based on the TwinsUK, LifeLines-DEEP, and UK biobank cohorts[11]. They identified a potential causal relationship between bacteria features (order and family levels) and human BMD measured by quantitative ultrasound of the heel[11], although no in vitro/in vivo experiments were

---

conducted to validate the findings. Other early studies were mainly limited by small sample sizes[5–10], and therefore were not able to identify specific bacterial species and/or associated functional mechanisms.

To identify significant gut bacteria to the species level, and the functional mechanisms by which they impact bone health, we performed a systematic integrative multi-omics analysis of the human genome, GM, and targeted metabolome [serum short chain fatty acids (SCFAs)] in 517 peri- and post-menopausal Chinese women. While many previous studies for OP have compared low/high BMD groups, the case-control design often reduces statistical power due to loss of information from choosing an artificial cut-off value for group selection[12]. To make full use of the whole samples, and thus bolster statistical power (compared with case-control analysis), we explored the association among GM, SCFA, and continuous BMD values. We identified several bacterial species, SCFAs, and functional pathways that significantly impacted BMD. We validated the major results in US white people and investigated the causal roles of individual bacterial species and SCFAs in regulating bone metabolism both in vitro in cultured cells and in vivo in mice. The results lay a foundation for potential prevention/intervention/treatment of OP through GM and metabolome alteration.

## Results

### Chinese study cohort characteristics

517 peri-/post-menopausal Chinese women were randomly recruited from Guangzhou City, China; 84% were post-menopausal and 16% peri-menopausal, based on years since menopause (YSM). Mean YSM and follicle stimulating hormone (FSH) levels were 1.96 years and 76.24 mIU/mL, respectively (Table 1). Age (mean ± standard deviation [SD], 52.85 ± 2.92 years) and body mass index (BMI) (22.97 ± 2.87 kg/m$^2$) were relatively homogeneous. Mean BMD of the lumbar spine (L1-L4), left total hip (HTOT, including femoral neck, trochanter, and inter-trochanter), femoral neck (FN), and ultra-distal radius and ulna (UD-RU) are shown in Table 1. Based on WHO criteria, 54.5% of the subjects had normal BMD (T-score ≥ −1), 38.5% osteopenia (−2.5 < T-score < −1), and 7% OP (T-score ≤ −2.5). Lifestyle factors (e.g., alcohol consumption, smoking, calcium supplementation, exercise) and socioeconomic status (e.g., education and family annual income) are also shown in Table 1. The associations between these factors and BMD/SCFAs are shown in Fig. S1. Stool DNA and blood samples were collected for metagenomic, genomic, and metabolomic analyses.

### Correlation between GM and BMD

We performed metagenomic shotgun sequencing on stool DNA samples (n = 499) and obtained ~7.35 Gbp of sequencing data per sample on average. The number of gut species increased with sample size, approaching a plateau (exceeded $1 \times 10^5$) with our study sample (Fig. 1a), indicating that our sample size was appropriate, large and reasonably powerful for this investigation. We identified 10,303 gut bacterial species at species level resolution by taxonomy annotation (ftp://ftp.ncbi.nlm.nih.gov/blast/db/FASTA/nr.gz). The three most common species were *Faecalibacterium prausnitzii*, *Bacteroides vulgatus*, and *Bacteroides fragilis* (Fig. 1b).

We used "vegan" package and Microbiome Regression-based Kernel Association Test (MiRKAT)[13] to calculate α-diversity (Shannon index) and β-diversity (optimal kernel, based on weighted UniFrac, unweighted UniFrac, and Bray-Curtis distance metrics), and evaluated their associations with BMD. GM biodiversity was negatively associated with forearm BMD, including ultra-distal radius (UD-R) BMD, ultra-distal ulna (UD-U) BMD, and UD-RU BMD (p-values < 0.05, q-values < 0.1, Table 2). Although no strong significant global association was found between GM biodiversity and BMD at L1-L4 or HTOT, we hypothesized that there may still be individual bacterial species contributing to these phenotypes. The GM biodiversity was calculated

**Table 1 | Characteristics of the Chinese study cohort**

| Phenotypes | Max | Min | Mean | SD |
|---|---|---|---|---|
| Age (years) | 64.59 | 41.47 | 52.85 | 2.92 |
| BMI (kg/m$^2$) | 33.73 | 16.42 | 22.97 | 2.87 |
| YSM (years) | 8.99 | 0.06 | 1.96 | 0.94 |
| FSH (mIU/mL) | 208.20 | 1.08 | 76.24 | 32.63 |
| L1-L4 BMD (g/cm$^2$) | 1.85 | 0.69 | 1.05 | 0.16 |
| HTOT BMD (g/cm$^2$) | 1.36 | 0.61 | 0.93 | 0.12 |
| FN BMD (g/cm$^2$) | 1.26 | 0.58 | 0.86 | 0.12 |
| UD-RU BMD (g/cm$^2$) | 0.61 | 0.22 | 0.40 | 0.06 |
|  | **None** | **Low** | **High** | |
| Calcium supplementation | 301/517 (58.2%) | 139/517 (26.9%) | 77/517 (14.9%) | |
| Exercise | 154/517 (29.8%) | 48/517 (9.3%) | 315/517 (60.9%) | |
| Family annual income | 139/517 (26.9%) | 221/517 (42.7%) | 157/517 (30.4%) | |
|  | **Yes** | **No** | | |
| Alcohol consumption | 146/517 (28.2%) | 371/517 (71.8%) | | |
| Smoking | 0 | 100% | | |
| Physically demanding jobs | 218/517 (42.2%) | 299/517 (57.8%) | | |
|  | **Less than high school** | **High school** | **College** | **Graduate** |
| Education | 153/517 (29.6%) | 153/517 (29.6%) | 195/517 (37.7%) | 16/517 (3.1%) |
|  | **Max** | **Min** | **Median** | |
| Caproic acid (µg/ml) | 0.671 | 0.017 | 0.049 | |
| Isovaleric acid (µg/ml) | 0.115 | 0.004 | 0.026 | |
| Butyric acid (µg/ml) | 0.354 | 0.030 | 0.074 | |
| Acetic acid (µg/ml) | 2.537 | 0.093 | 0.558 | |
| Isobutyric acid (µg/ml) | 0.193 | 0.003 | 0.037 | |
| Valeric acid (µg/ml) | 0.124 | 0.001 | 0.010 | |

Note: Calcium supplementation: None - never had calcium supplementation; Low - occasional calcium supplementation; High - daily calcium supplementation. Exercise: None - do not have exercise habit; Low - exercise <2.5 h per week; High - exercise >2.5 h per week. Family annual income: None - low family income; Low - lower than the local average income but higher than the low family income; High - higher than the local average income.

*YSM* years since menopause, *FSH* follicle stimulating hormone, *BMI* body mass index, *BMD* bone mineral density, *L1-L4* lumbar spine, *HTOT* left total hip, *FN* femoral neck, *UD-RU* ultra-distal radius and ulna.

using counts of individual species[14], so it is plausible that combinations of different species may lead to the same biodiversity. Therefore, we proceeded with further analysis for identification of individual bacterial species associated with BMD.

We transformed the relative abundances using the centered log ratio (CLR) transformation and applied constrained linear regression analysis for compositional data (e.g., relative abundance of bacterial species)[15]. Several species were identified to be significantly associated with BMD at the various skeletal sites (Table S1). In particular, *Bacteroides vulgatus* was found to be significantly associated with L1-L4 BMD (β = −0.027, p-value = 0.032, Fig. 1c). The effect size of *B. vulgatus* had a similar magnitude of association impact to other common BMD-related covariates, such as BMI, YSM, and exercise (Table S1), which suggests that this species may play an important role in BMD.

After functional annotation and obtaining Kyoto Encyclopedia of Genes and Genomes (KEGG) modules, we performed partial Spearman correlation analysis to identify relationships between GM functional capacity and BMD variation. Three functional modules were negatively associated with L1-L4 BMD and four modules were negatively

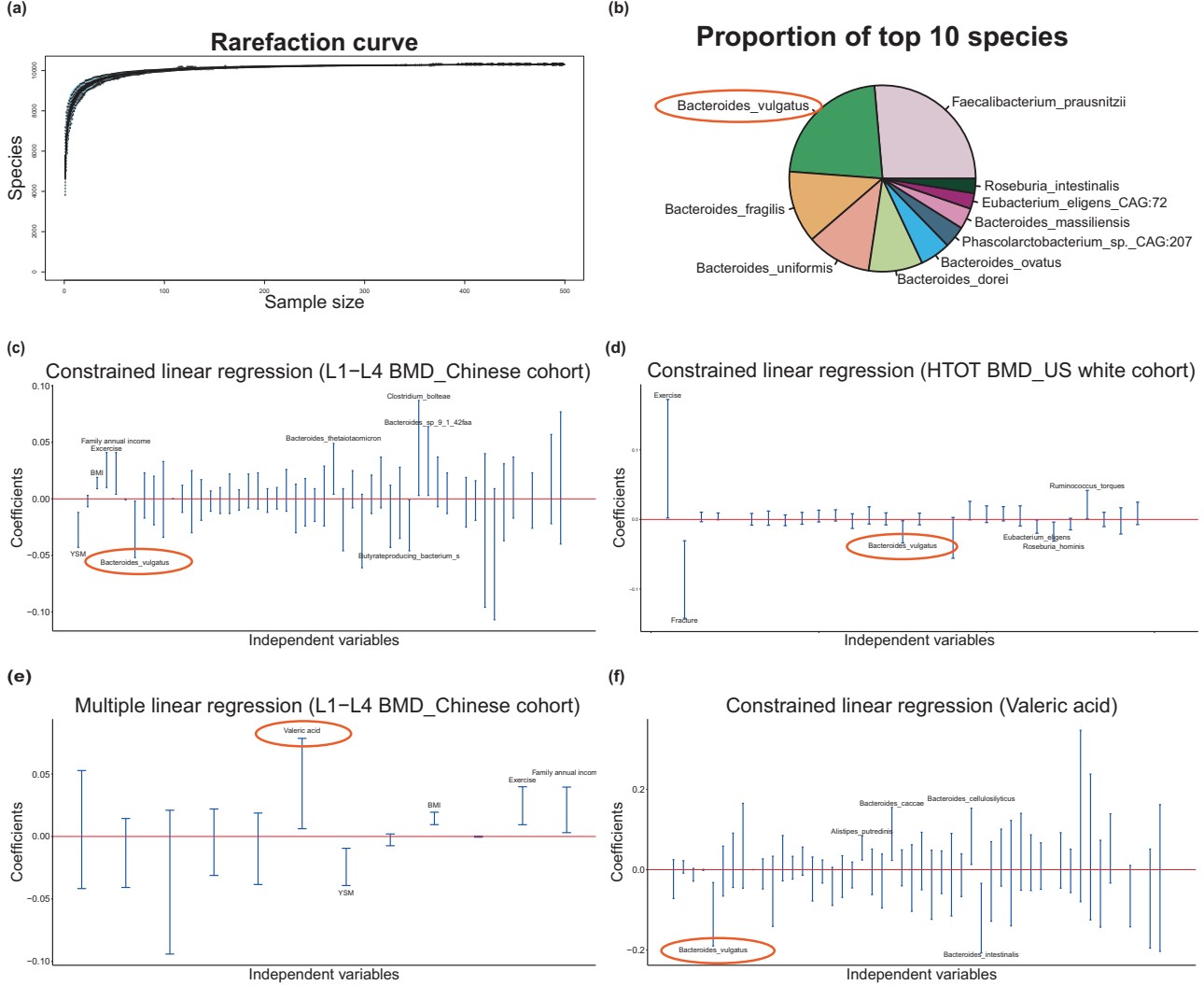

**Fig. 1 | Bacterial species composition and associations among bacterial speices, SCFAs and BMD. a** Rarefaction curve: the number of bacterial species (*Y*-axis) was plotted against the number of samples (*X*-axis, *n* = 499). **b** Bacterial species composition in the study cohort composed of top 10 bacterial species. **c–f** Bacterial species/SCFAs related to BMD/VA in the Chinese cohort and US cohort by constrained linear regression analysis or multiple linear regression analysis. Regression coefficients (*Y*-axis) were plotted against independent variables (*X*-axis). Species/SCFAs and covariates were considered as independent variables and represented by the blue lines. The length of the blue line indicates the 95% confidence interval (CI). **c** L1-L4 BMD-related species of Chinese cohort (*n* = 499); **d** HTOT BMD-related speices of the US white cohort (*n* = 59); **e** L1-L4 BMD-related SCFAs of of Chinese cohort (*n* = 500); **f** Valeric acid-related species of Chinese cohort (*n* = 499). Source data are provided as a Source Data file. SCFAs short chain fatty acids, BMD bone mineral density, L1-L4 lumbar spine, HTOT left total hip, YSM years since menopause, BMI body mass index.

associated with UD-RU BMD (γ's < −0.10, *p*-values < 0.01, *q*-values < 0.25, Table 3). Since the *q*-values are >0.050, we used a stringent threshold for determining significance (*p*-value < 0.010). All these modules involve pathways whereby GM produces metabolites from ingested foods, suggesting that the produced metabolites by GM may influence BMD.

**SCFAs significantly associated with BMD**
Since the GM may affect host health through SCFAs[16], we analyzed serum SCFA levels (*n* = 500) by targeted metabolomics for six microbiota derived metabolites (Table 1). Based on multiple linear regression analysis, valeric acid (VA) was positively associated with L1-L4 BMD (*β* = 0.044, *p*-value = 0.017, Fig. 1e). The remaining SCFAs (including caproic acid, isovaleric acid, butyric acid, acetic acid, and isobutyric acid) were not significantly associated with BMD at any skeletal sites in this sample (Table S2).

The constrained linear regression analysis revealed that five bacterial species were significantly associated with VA (Fig. 1f and

Table S1). In particular, *B. vulgatus* was negatively associated with VA (*β* = −0.111, *p*-value = 0.006). Since *B. vulgatus* was significantly associated with both L1-L4 BMD (Fig. 1c) and VA (Fig. 1f), we focused on this species for the subsequent analyses.

**Potential causality between *B. vulgatus* and valeric acid**
Since VA is produced by *Oscillibacter valericigenes*[17] and *Megasphaera elsdenii*[18] (both of which are potential probiotics[19,20]), we calculated the Spearman correlation between *B. vulgatus* and these two VA-producing microbes (probiotics) and found that the relative abundance of *B. vulgatus* was negatively associated with both the *Oscillibacter valericigenes* (γ = −0.41, *p*-value = 2.2 × 10⁻¹⁶) and *Megasphaera elsdenii* (γ = −0.17, *p*-value = 1.2 × 10⁻⁴) in humans.

We further performed whole genome sequencing (WGS) and GWAS for the Chinese cohort (*n* = 500) to obtain single nucleotide polymorphisms (SNPs) as instrumental variables (IVs) for subsequent MR analysis. After removing duplicate reads, the average sequencing depth on the whole genome excluding gap regions was 17.69-fold. On

**Table 2 | Correlation between GM biodiversity and BMD variation**

| Phenotypes | Shannon index | | | Optimal Kernel | |
|---|---|---|---|---|---|
| | partial γ | p-value | q-value | p-value of MiRKAT | q-value of MiRKAT |
| **UD-R BMD** | −0.081 | 0.072 | 0.144 | 0.005 | 0.028 |
| **UD-U BMD** | −0.098 | 0.029 | 0.093 | 0.014 | 0.028 |
| **UD-RU BMD** | −0.097 | 0.031 | 0.093 | 0.010 | 0.028 |
| HTOT BMD | −0.043 | 0.344 | 0.516 | 0.812 | 0.812 |
| FN BMD | 0.004 | 0.931 | 0.931 | 0.733 | 0.812 |
| L1-L4 BMD | 0.031 | 0.493 | 0.592 | 0.475 | 0.713 |

Note: partial γ - coefficient for the correlation between the Shannon index and BMD variation with partial Spearman correlation analysis; p-value - p-value of the correlation coefficient γ (two-sided); q-value - false discovery rate of the partial Spearman correlation analysis; Optimal Kernel - optimal kernel based on weighted and unweighted UniFrac distance matrices and Bray-Curtis distance; p-value of MiRKAT - p-value of the association between GM biodiversity and BMD variation with MiRKAT (two-sided); q-value of MiRKAT - false discovery rate of the MiRKAT analyses.
Bolded BMDs were the nominally significant ones associated with GM biodiversity (p-values < 0.05).

**Table 3 | Correlation between GM functional capacity and BMD variation**

| Module ID | γ | p-value | q-value | Module names | Phenotypes |
|---|---|---|---|---|---|
| M00002 | −0.122 | 0.007 | 0.234 | Glycolysis | L1-L4 BMD |
| M00048 | −0.119 | 0.008 | 0.234 | IMP biosynthesis | L1-L4 BMD |
| M00855 | −0.119 | 0.008 | 0.234 | Glycogen degradation | L1-L4 BMD |
| M00002 | −0.118 | 0.009 | 0.153 | Glycolysis | UD-RU BMD |
| M00003 | −0.119 | 0.008 | 0.153 | Gluconeogenesis | UD-RU BMD |
| M00050 | −0.143 | 0.002 | 0.079 | GTP biosynthesis | UD-RU BMD |
| M00727 | −0.135 | 0.003 | 0.079 | CAMP resistance | UD-RU BMD |

Note: γ - coefficient for the partial Spearman correlation between GM functional module and phenotype; p-value - p-value of the coefficient in partial Spearman correlation analysis (two-sided); q-value - p-value adjusted by false discovery rate; Glycolysis - core module involving three-carbon compounds; IMP biosynthesis - PRPP + glutamine → IMP; Glycogen degradation - glycogen → glucose-6P; Gluconeogenesis - oxaloacetate → fructose-6P; GTP biosynthesis - IMP → GDP, GTP; CAMP resistance - N-acetylmuramoyl-L-alanine amidase AmiA and AmiC. We only tested the modules whose relative abundance > 0.10%.
GM gut microbiota, BMD bone mineral density, L1-L4 lumbar spine, UD-RU ultra-distal radius and ulna, PRPP phosphoribosylpyrophosphate, IMP inosine monophosphate, GDP guanosine 5′-diphosphate, GTP guanosine 5′-triphosphate, CAMP cationic antimicrobial peptide.

average, per sequenced individual, 98.48% of the whole genome excluding gap regions were covered by at least 1× coverage, 96.85% had at least 4× coverage and 89.52% had at least 10× coverage. After quality control (QC), there are 9,120,095 SNPs included. Based on the QQ plot of association statistics, we observed that potential confounding due to potential population structure was minimal (genomic inflation factor_Bacteroides vulgatus = 1.016, genomic inflation factor_valeric acid = 1.028, Fig. S2).

The MR approach[21] was then used to assess the potential causality of B. vulgatus (as exposure) on VA (as outcome) in humans, using SNPs derived from whole genome sequencing for the same study subjects as IVs. Overall, we obtained 15 linkage disequilibrium (LD)-independent SNPs that were associated with B. vulgatus and were included in the MR analysis. The mean value of the F-statistics is 30.837 (F-statistic > 10), indicating that there is no weak instrument bias (Table S3). Most MR analyses, including those using the weighted median method, maximum likelihood estimation (MaxLik), and inverse-variance weighted approach all indicated that B. vulgatus may causally down-regulate VA (β's < −0.07, p-values < 0.05, Table 4). In addition, the intercept from the MR-Egger regression model was not significant (p-value = 0.517),

**Table 4 | Potential causality of *Bacteroides vulgatus* on VA with MR approach**

| MR methods | β | Standard error | p-value |
|---|---|---|---|
| Weighted median method | −0.116 | 0.038 | 0.002 |
| MaxLik | −0.075 | 0.033 | 0.022 |
| IVW | −0.071 | 0.032 | 0.025 |
| MR-Egger | −0.031 | 0.069 | 0.655 |
| MR-Egger (intercept) | −0.023 | 0.035 | 0.517 |

Note: β - regression coefficient for the association between *Bacteroides vulgatus* (as exposure) and valeric acid (as outcome) with various MR methods; p-value - p-value of the regression coefficient.
VA valeric acid, MR Mendelian randomization, MaxLik maximum likelihood estimation, IVW inverse-variance weighting.

indicating that there are no horizontal pleiotropic effects influencing the causal relationship between B. vulgatus and VA levels. Therefore, the selected SNPs influence VA only through B. vulgatus, and are unlikely to do so through other independent pathways.

### B. vulgatus/BMD associations in US white people
To assess the robustness of our major findings in the Chinese cohort, we tested associations between B. vulgatus and human BMD in an independent cohort of 59 post-menopausal US white females, aged ≥60 years, in New Orleans, Louisiana (Table S4). Mean (SD) age and BMI were 66.98 (5.65) years and 27.84 (8.50) kg/m², respectively. Mean BMD for L1-L4, HTOT, FN, UD-R, and UD-U were 0.92, 0.80, 0.67, 0.37, and 0.28 g/cm², respectively. Lifestyle information is also shown in Table S4. We detected an association between B. vulgatus and HTOT BMD (β = −0.018, p-value = 0.029, Fig. 1d), supporting B. vulgatus's association with BMD in ethnically distinct female populations. Regression coefficients and their corresponding p-values of BMD-associated gut bacterial species/covariates at various skeletal sites in the US sample are shown in Table S5. However, we were not able to replicate the effect of B. vulgatus on L1-L4 BMD specifically.

To evaluate the overall effects of B. vulgatus on BMD, we further performed the one-sided sign test[22] based on the regression coefficients of B. vulgatus with each of the individual skeletal sites (Table S6). The results demonstrated that there were significant overall negative associations between B. vulgatus and BMD in both the Chinese and US cohorts (Chinese sample p-value = $4.8 \times 10^{-7}$, US sample p-value = $9.8 \times 10^{-4}$).

### In vivo and in vitro validation
**Effects of B. vulgatus on VA levels and bone resorption in vivo.** After gavage with B. vulgatus/normal saline (NS) every other day for eight weeks, we observed poorer bone micro-structure in the OVX-NS mice, versus the sham-operated-NS mice (including lower trabecular number [Tb.N] and bone volume/tissue volume [BV/TV] but higher trabecular separation [Tb.Sp], p-values < 0.001, q-values < 0.001, Fig. 2f–i), indicating bone loss in the OVX-mouse model. Similar results can be identified in the comparisons of the OVX-B. vulgatus-mice, versus sham-B. vulgatus-mice. In the OVX-B. vulgatus group (versus the OVX-NS group), we observed lower trabecular thickness [Tb.Th] and BV/TV (p-values < 0.001, q-values < 0.001, Fig. 2g, i), as well as a trend of higher Tb.Sp (p-value = 0.159, q-value = 0.212, Fig. 2h) and a trend of lower percentages of mineralized volume (quantified by percentages of mineralized area in Von Kossa staining, p-value = 0.104, q-value = 0.139, Fig. 2j). We also observed an increased number of osteoclasts in the lumbar vertebral body (p-value = 0.010, q-value = 0.020, Fig. 2l), higher bone resorption levels (measured by type I collagen serum C-telopeptide [CTX-I], p-value = 0.033, q-value = 0.035, Fig. 2r). In addition, we observed higher relative abundance of B. vulgatus (p-value = 0.007, q-value = 0.014, Fig. 2m)

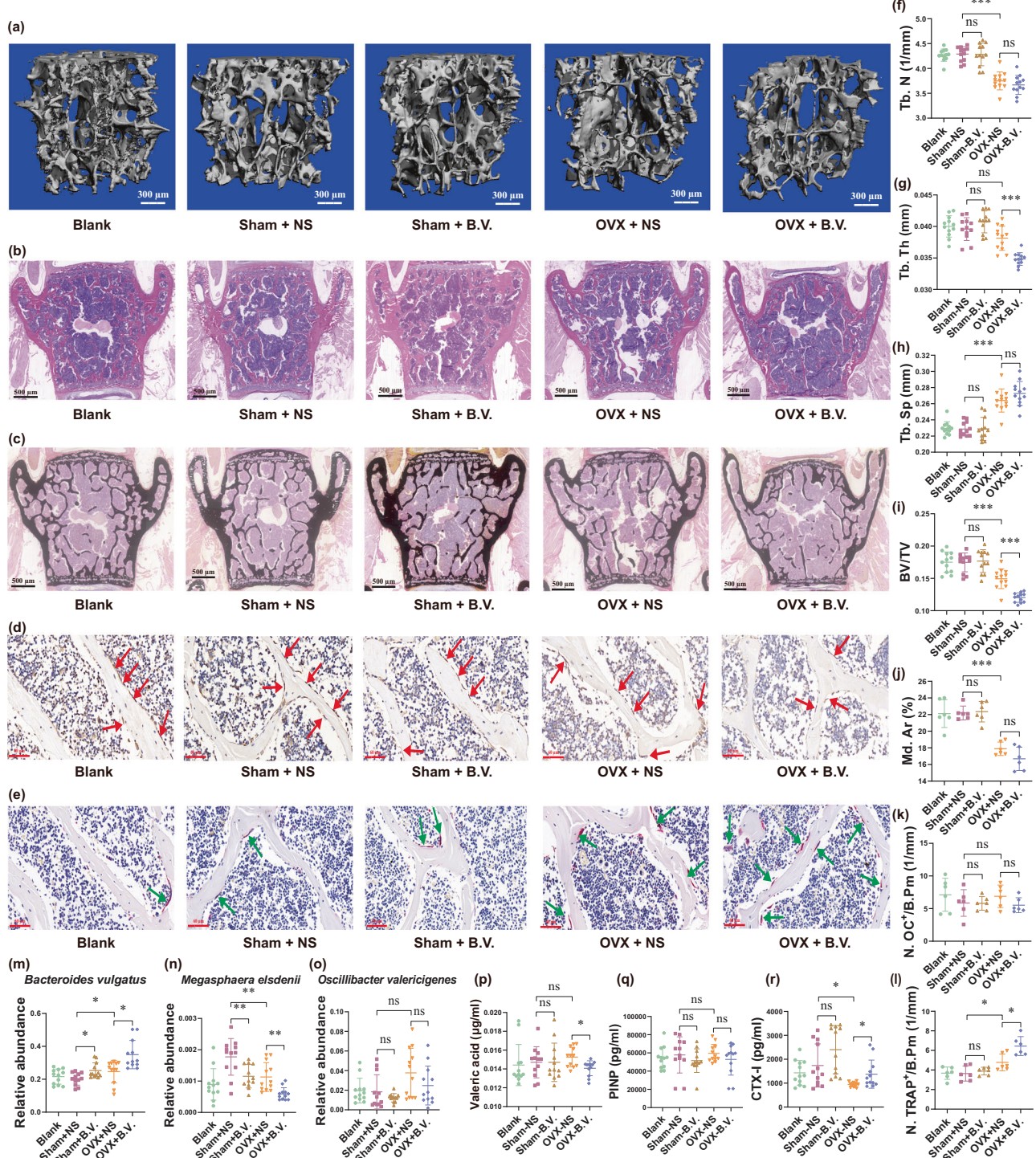

**Fig. 2 | *Bacteroides vulgatus* regulates bone-associated phenotypes of mice in vivo.** Changes in various bone-associated phenotypes in mice after gavage with *B. vulgatus* or NS for 8 weeks after OVX, compared with sham (non-OVX) and blank (non-OVX and no oral gavage) female mice (n_mice = 12/group): **a**–**e** representative microCT/HE staining/Von Kossa staining/IHC-OC staining/TRAP staining images of the 5th lumbar vertebral body. Arrows point to osteocalcin positive cells (**d**) and osteoclasts positive cells (**e**). **f**–**i** Quantitative indices of trabecular bone volume and structure, including Tb.N, Tb.Th, Tb.Sp, and BV/TV (n_mice = 12/gourp). **j** Mineralized volumes of mice bone are quantified by percentages of mineralized area in Von Kossa staining (n_mice = 6/group). **k**, **l** Quantitative data of osteocalcin positive cells and osteoclasts, respectively (n_mice = 6/group). **m**–**o** Relative abundance of *B. vulgatus* and valeric acid-producing species (*Megasphaera elsdenii*

and *Oscillibacter valericigenes*) (n_mice = 12/group). **p**–**r** serum levels of valeric acid, PINP concentrations, and CTX-I concentrations (n_mice = 12/group). microCT - micro-computed tomography, B.V. *Bacteroides vulgatus*, BV/TV bone volume/tissue volume, CTX-I C-telopeptide of type I collagen, HE hematoxylin-eosin, IHC-OC immunohistochemistry-osteocalcin, Md. Ar mineralized area, NS normal saline, N.OC$^+$/B.Pm osteocalcin positive cells number per analyzed bone perimeter, N.TRAP$^+$/B.Pm TRAP-stained osteoclast number per analyzed bone perimeter, OVX ovariectomized, PINP procollagen I N-terminal propeptide. Tb.N trabecular number, Tb.Sp trabecular separation, Tb.Th trabecular thickness. Data are presented as mean values with SD. ns indicates non-significant, • indicates *q*-value < 0.05, •• indicates *q*-value < 0.01, ••• indicates *q*-value < 0.001. Source data are provided as a Source Data file.

while lower/a trend of lower relative abundance of VA-producing microbes (*Megasphaera elsdenii*, *p*-value = 0.001, *q*-value = 0.001, Fig. 2n; *Oscillibacter valericigenes*, *p*-value = 0.160, *q*-value = 0.213, Fig. 2o) and lower serum levels of VA (*p*-value = 0.008, *q*-value = 0.034, Fig. 2p). However, we did not observe significant changes of bone formation related phenotypes (including Tb.N, number of osteoblasts in the lumbar vertebral body, and serum levels of procollagen I N-terminal propeptide [PINP], *p*-values > 0.05, Fig. 2f, k and q) between the OVX-*B. vulgatus* group and OVX-NS group. Meanwhile, the bone related phenotypes are non-significant between the sham groups of mice (sham-*B. vulgatus* group vs sham-NS group, *p*-values > 0.05, Fig. 2). These results suggest that *B. vulgatus* suppresses VA and promotes bone resorption, leading to deteriorated bone architecture in the OVX mice model; such effects were not detected in the sham model.

**Effects of VA on bone metabolism in vivo**. To investigate the direct causal-effect of VA on bone formation/resorption, we performed another in vivo experiment in which 8-week-old female C57BL/6J mice (*n* = 12/group) were fed VA (1.5 mM) in drinking water (free-drinking), compared with normal drinking water without VA treatment (0 mM).

After eight weeks of intervention, we observed poorer bone microstructure in the OVX-normal drinking water mice, versus sham-operated-normal drinking water mice (*p*-values < 0.001, *q*-values < 0.001, Fig. 3f–i), indicating bone loss in the OVX-mouse model. Similar results can be identified in the comparisons between the OVX-VA-mice versus sham-VA-mice. In the OVX-VA group (versus the OVX-normal drinking water group), we observed higher Tb.Th (*p*-value = 0.004, *q*-value = 0.005, Fig. 3g) and BV/TV (*p*-value = 0.009, *q*-value = 0.012, Fig. 3i), as well as a trend of higher percentages of mineralized volume (quantified by percentages of mineralized area in Von Kossa staining, *p*-value = 0.090, *q*-value = 0.120, Fig. 3j), and a trend of lower serum levels of CTX-I (*p*-value = 0.037, *q*-value = 0.106, Fig. 3o). We also observed decreased numbers of osteoclasts in the lumbar vertebral body (*p*-value = 0.005, *q*-value = 0.016, Fig. 3l). In addition, we observed higher serum levels of VA (*p*-value = 0.004, *q*-value = 0.007, Fig. 3m). However, we did not observe significant changes of bone formation related phenotypes (including Tb.N, number of osteoblasts in the lumbar vertebral body, and serum levels of PINP, *p*-values > 0.05, Fig. 3f, k and n) between the OVX-VA group and the OVX-normal drinking water group. These results suggest that VA suppressed bone resorption, preventing deteriorated bone architecture in the OVX model.

**Effects of VA on osteoclast-like cell and osteoblast in vitro**. Since the results suggest that *B. vulgatus* affects BMD by down-regulating VA production, we investigated the effects of VA on osteoclast-like cell and osteoblast differentiation in vitro. Murine macrophages (RAW264.7) and pre-osteoblasts (MC3T3-E1) were differentiated with/without VA treatment. We tested various VA concentrations (1, 0.1, and 0.01 mM, Fig. S3) at first and found that 1 mM is the most suitable concentration for cell differentiation. After inducing osteoclast-like cell differentiation of RAW264.7 cells with receptor activator of nuclear factor-κB ligand (RANKL, 50 ng/mL), 5 days treatment with VA (1 mM) significantly decreased the number of mature osteoclast-like cells vs. controls (Fig. 4a, two-sample *t*-test *p*-value < 0.001), indicating that VA significantly inhibited osteoclast-like cell differentiation. Treatment with VA (1 mM) for 14 days also significantly increased differentiation of MC3T3-E1 into osteoblasts with a higher Alkaline phosphatase (ALP) activity (top panel of Fig. 4b, *p*-value < 0.001), and increased mineralization of the extracellular matrix (bottom panel of Fig. 4b, *q*-value = 0.007).

Since inflammatory pathways play important roles in bone metabolism[23] and GM may potentially influence bone metabolism through its effects on inflammation[24], we measured expression levels of several critical inflammatory genes (e.g., phosphorated RELA [p-

RELA], p-CHUK, p-IKBKB, p-NFKBIA, TNF and IL10) in cultured osteoclast-like cells and osteoblasts. After treating osteoclast-like cells and osteoblasts with VA for 5 and 14 days respectively, we observed decreased expression of p-RELA, along with increased expression of p-NFKBIA, p-CHUK/IKBKB, and mRNA levels of IL10 and TNF in VA-treated RAW264.7 cells. We also observed increased expression of IL10 mRNA levels and p-NFKBIA, and decreased expression of p-RELA in VA-treated MC3T3-E1 cells (Fig. 4c–f, *p*-values < 0.05). RELA is a subunit of NF-κB which plays a critical role in the pathogenesis of multiple chronic inflammatory diseases[25], while IL10 is an anti-inflammatory cytokine[26]. Thus, VA inhibited inflammatory responses of osteoclast-like cells and osteoblasts. This may partially explain its positive association with BMD since inflammation promotes bone loss[23], and NF-κB activity can inhibit osteoblast functions[27].

## Discussion

We found, in human studies, that: (1) GM biodiversity and several bacterial metabolic pathways were negatively associated with BMD; (2) several individual bacterial species (especially *B. vulgatus*) were significantly associated with BMD in ethnically distinct populations; (3) serum VA levels were positively associated with L1-L4 BMD; (4) *B. vulgatus* was negatively associated with serum VA levels and may causally suppress VA levels, presumably by inhibiting growth of VA-producing bacteria within the gut.

Subsequent in vivo and in vitro studies found that: (1) VA levels were decreased, while bone resorption was increased, in OVX-mice fed *B. vulgatus*; (2) VA suppressed bone resorption of OVX-mice; (3) VA suppressed maturation of osteoclast-like cells, and promoted maturation of osteoblasts and extracellular matrix mineralization by osteoblasts in vitro; (4) VA treatment of osteoclast-like cells and osteoblasts in vitro suppressed RELA protein production (pro-inflammatory), and enhanced IL10 mRNA expression (anti-inflammatory). Thus, *B. vulgatus* appears to decrease BMD by reducing VA production within the gut, resulting in enhanced inflammation and osteoclast activity, and reduced osteoblast activity (Fig. 5).

The negative association between GM biodiversity and forearm BMD in our discovery sample is consistent with previous reports. Wang et al.[8] and Das et al.[6] found elevated/a trend of higher diversity in OP and osteopenia groups compared with controls. It has also been shown that germ-free mice (non-existent GM biodiversity) had higher bone mass than mice with normal GM[4], and that antibiotic treatment of mice early in life reduced GM biodiversity and increased bone size[28]. Although GM biodiversity is generally believed to be beneficial[29], the above results suggest that the effects of GM biodiversity on health may vary with distinct phenotypic traits, environmental factors (e.g. diet and exercise) and individual species composition within GM.

In this study, *B. vulgatus* is one of the individual bacterial species found to be associated with BMD. *B. vulgatus* is one of the most abundant bacterial species in the guts of mammals (including mice and humans); it belongs to the *Bacteroides* genus[30], one of the most abundant genera in the human microbiome[30]. *B. vulgatus* can trigger pro-inflammatory NF-κB signaling pathways[31] which are associated with bone remodeling[27]. We did not observe the effect of *B. vulgatus* on the sham-operated groups of mice, which indicates that the *B. vulgatus* may have distinct effects in eugonadic mice and estrogen deficient mice. Since menopause is a key period of change for women's health (with, e.g., increased risk of cardiovascular and metabolic diseases)[32] and *B. vulgatus* is an opportunistic pathogen[33], we speculate that the *B. vulgatus* produces a harmful effect on BMD mainly under estrogen deficient conditions. This speculation is consistent with reports that *B. vulgatus* is elevated with polycystic ovary syndrome-associated ovarian dysfunction[34].

Although the discovery cohort demonstrated an association of *B. vulgatus* with spine BMD while the replication cohort indicated an

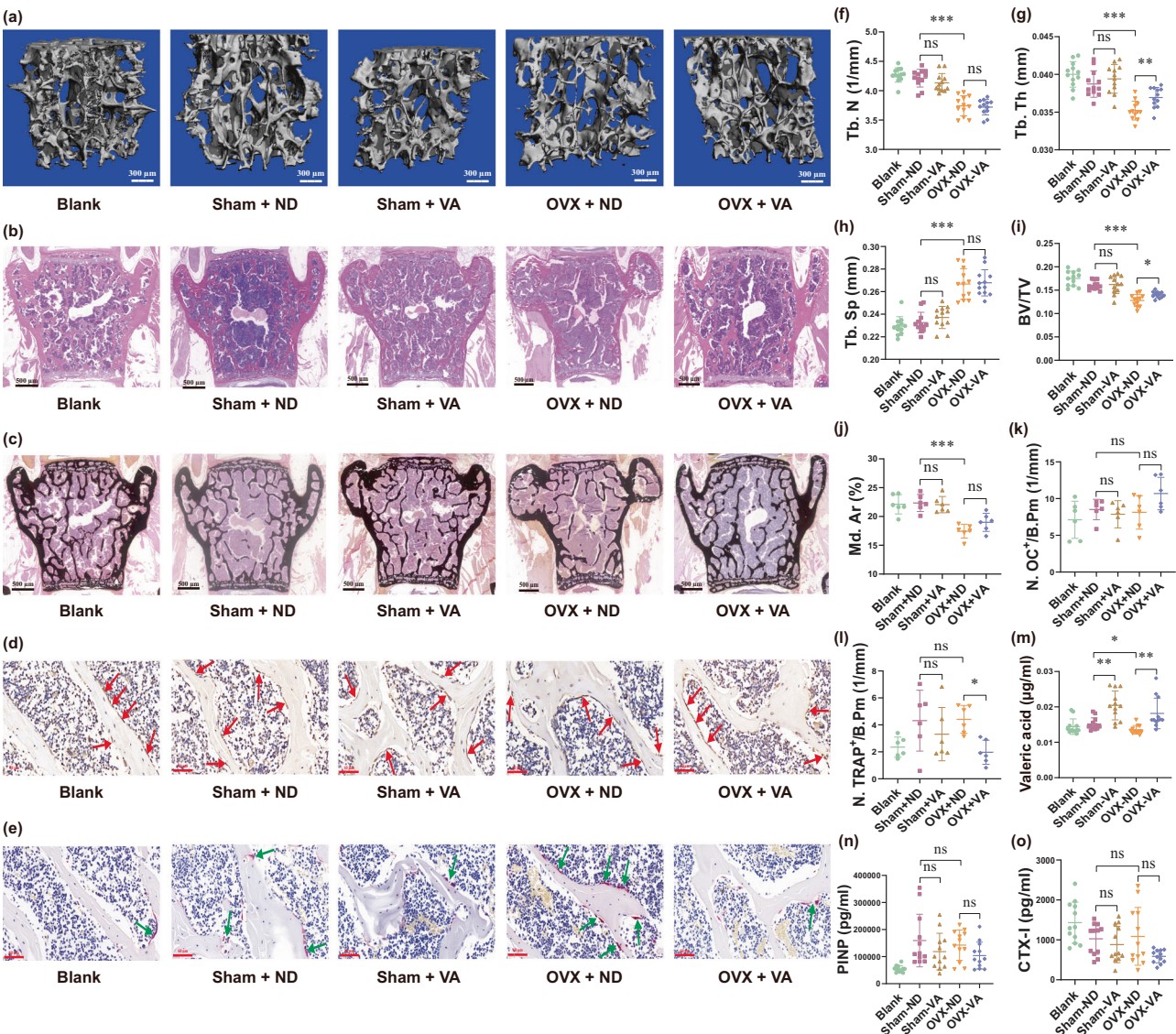

**Fig. 3 | Valeric acid influences bone-associated phenotypes of mice in vivo.** Changes in various bone-associated phenotypes in mice after free-drinking water with/without valeric acid for 8 weeks after OVX, compared with sham (non-OVX) and blank (non-OVX and no oral gavage) female mice (n_mice = 12/group): **a**–**e** representative microCT/HE staining/Von Kossa staining/IHC-OC staining/TRAP staining images of the 5th lumbar vertebral body. Arrows point to osteocalcin positive cells (**d**) and osteoclasts positive cells (**e**). **f**–**i** Quantitative indices of trabecular bone volume and structure, including Tb.N, Tb.Th, Tb.Sp, and BV/TV (n_mice = 12/group). **j** mineralized volumes of mice bone are quantified by percentages of mineralized area in Von Kossa staining (n_mice = 6/group). **k**, **l** Quantitative data of osteocalcin positive cells and osteoclasts, respectively (n_mice = 6/group). **m**–**o** Serum levels of valeric acid, PINP concentrations, and CTX-I concentrations (n_mice = 12/group). Data are presented as mean values with SD. ns indicates non-significant, ∗ indicates $q$-value < 0.05, ∗∗ indicates $q$-value < 0.01, ∗∗∗ indicates $q$-value < 0.001. Source data are provided as a Source Data file. VA valeric acid, ND normal drinking water without valeric acid treatment.

association with hip BMD, it is not unusual, and actually commonly observed in epidemiology studies, that association results for BMD may differ for different skeletal sites. This is particularly true and widely observed for genetic epidemiology studies[35]. A prominent example is the association of the *WNT16* gene (a most robust associated gene with BMD identified in the field). *WNT16* was associated with hip BMD in European descent and Asian populations in some studies [e.g.,[36]], with lumbar spine BMD in European, Hispanic-American, and African-Americans in other studies [e.g.,[37]], and with forearm BMD in Europeans in other studies [e.g.,[38]]. It is well accepted that BMDs of different sites are correlated and can, together or individually, reflect general bone health[39]. In addition, the rigorous one-sided sign test results also clearly supported that overall there were significant negative associations between *B. vulgatus* and BMD in both the Chinese and US cohorts. Furthermore, the association and partial

validation between *B. vulgatus* and BMD in the human cohorts was primarily for discovery relative to informing the follow-up functional validation experiments. Following the discoveries based on association analyses in humans, we subsequently validated the results by various in vivo and in vitro experiments. Taken together, the collective results provide strong evidence for the significance of *B. vulgatus* on BMD and bone metabolism.

To determine how GM influences BMD, we examined relationships between GM functional capacity and BMD variation. Most GM functional modules that were negatively associated with BMD (Table 3) involved GM metabolism. In contrast to human metabolism, these BMD-related GM metabolism modules are essential for bacterial activity. Specifically, M00002, M00003 and M00855 involved glycolytic pathways or gluconeogenesis, both of which affect glucose metabolism, essential for bacterial cellular energy[40]. M00048 and

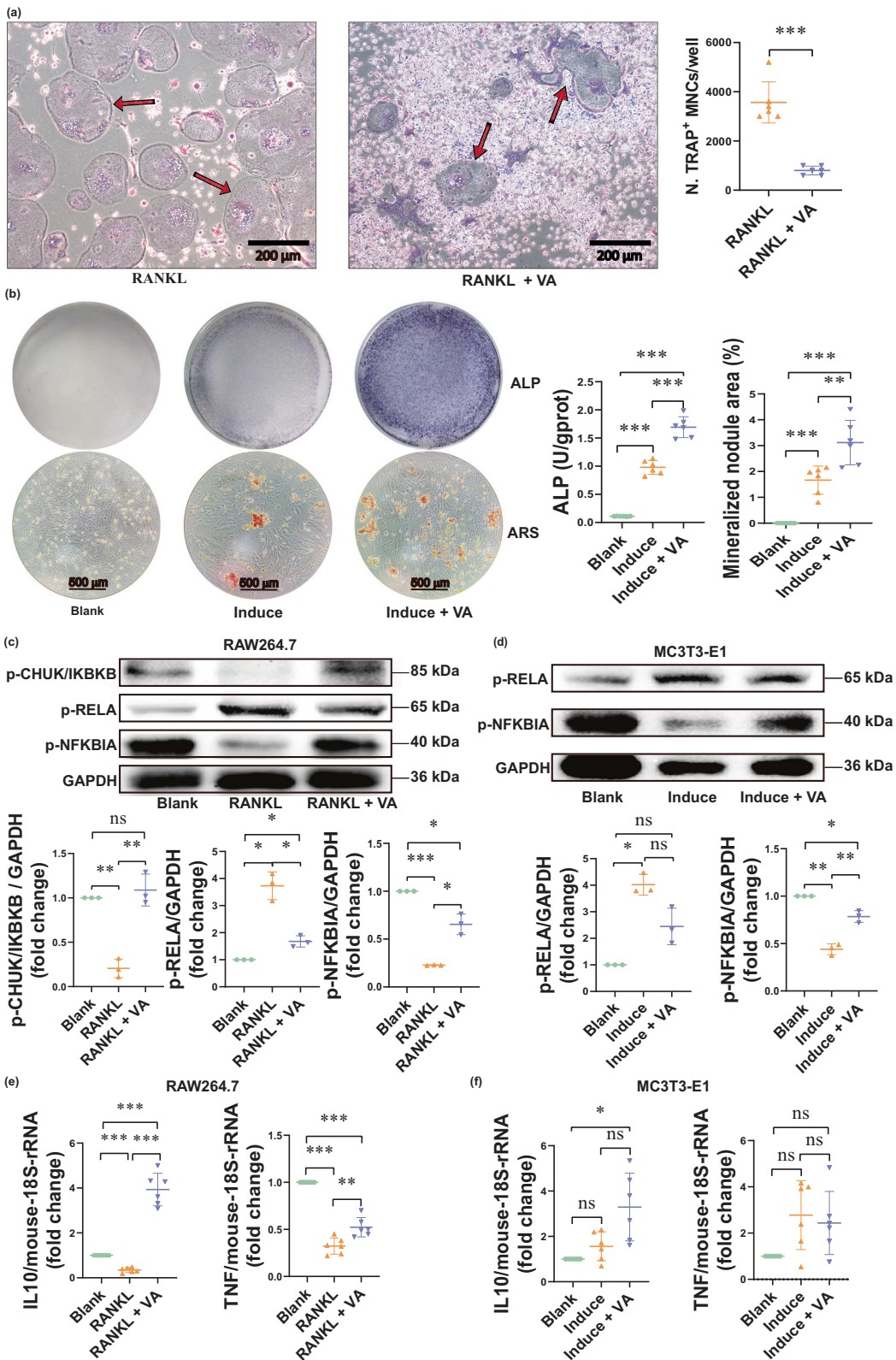

M00050 both involve nucleic acid synthesis required for bacterial growth[41]. These collective findings demonstrate that human bone health is affected by GM, and that specific bacterial biosynthetic/metabolic pathways contribute to these effects.

Since several GM functional modules were associated with BMD, and microbial metabolites contribute to host-microbiome interactions[42], we explored relationships between BMD and serum SCFAs. SCFAs are produced exclusively by GM, absorbed in the colon, and function as critical signaling molecules between the host and GM[16]. Butyric acid is the most widely studied SCFA for bone health and has been reported to inhibit osteoclast differentiation and bone resorption in mice[16], although we did not identify a significant association between butyric acid and BMD in this study. We did, however, demonstrate an important role for VA. VA has been reported to inhibit

**Fig. 4 | Valeric acid influences osteoclast-like cell and osteoblast differentiation in vitro.** Effects of valeric acid (VA) on osteoclast-like cell differentiation of RAW264.7 cells (induced by receptor activator of nuclear factor-κB ligand [RANKL]), osteoblast differentiation of MC3T3-E1 cells. Two-sample t-test was performed to identify the difference between groups. False discovery rate (q-value) was calculated for multiple testing correction on the p-values (two-sided): **a** Microscopic images of tartrate-resistant acid phosphatase (TRAP) staining of osteoclast-like cells induced from RAW264.7 cells after 5 days of osteoclastogenesis with/without VA treatment (n = 6). VA significantly decreased the number of mature osteoclast-like cells (two-sample two-sided t-test p-value < 0.001). Arrows indicate TRAP-positive multinucleated cells (TRAP⁺MNCs). **b** Alkaline phosphatase (ALP) staining for osteoblast differentiation and alizarin red S (ARS) staining for extracellular matrix mineralization by MC3T3-E1 cells after 14 days of osteo-blastogenesis with/without VA treatment (n = 6). VA significantly increased ALP activity (q-value < 0.001) and mineralization of the extracellular matrix (q-value = 0.007); dot plots show ALP activity (top panel) and quantification of ARS staining (bottom panel). **c** Western blot of p-RELA, p-NFKBIA, p-CHUK/IKBKB proteins in osteoclast-like cells with/without VA treatment (n = 3). VA significantly increased expression of p-CHUK/IKBKB (q-value = 0.009) and p-NFKBIA (q-value = 0.031), and decreased expression of p-RELA (q-value = 0.017). **d** Western blot of p-RELA and p-NFKBIA proteins in osteoblasts with/without VA treatment (n = 3). VA significantly increased expression of p-NFKBIA (q-value = 0.005) and caused a trend of decreased expression of p-RELA (p-value = 0.038, q-value = 0.057). **e** IL10 and TNF mRNA expressions in osteoclast-like cells with/without VA treatment (n = 6). VA significantly increased mRNA levels of IL10 and TNF (q-values < 0.01). **f** IL10 and TNF mRNA expressions in osteoblasts with/without VA treatment (n = 6). VA caused a trend of increased mRNA levels of IL10 (p-value = 0.036, q-value = 0.053). The "induce" means the MC3T3-E1 cells were induced into osteoblasts by osteoblastogenic medium without VA treatment. Data are presented as mean values with SD. • indicates q-value < 0.05, •• means q-value < 0.01, ••• indicates q-value < 0.001. Source data are provided as a Source Data file.

histone deacetylase (HDAC), an enzyme that is important for epigenomic regulation of gene expression[43]. HDACs are implicated in the pathogenesis of a number of diseases (e.g., cancer, colitis, cardiovascular disease and neurodegeneration), and HDAC inhibitors are considered as potential therapeutic agents[43]. In irradiated mice, VA can protect hematogenic organs, improve gastrointestinal tract function and enhance intestinal epithelial integrity to elevate survival rate[44]. In the current study we, for the first time, provide evidence suggesting that VA could potentially be used to treat OP. We found that VA was positively associated with BMD in humans, inhibited bone resorption in OVX-mice, and inhibited differentiation of osteoclast-like cells and promoted differentiation and mineralization of osteoblasts, in vitro. These collective results support a protective effect for VA on BMD.

Through further experiments, we revealed that the protective effect of VA is partially attributable to decreased osteoclast activity and increased osteoblast activity mediated by increased IL10 expression (anti-inflammatory) and inhibition of p-RELA expression (pro-inflammatory), along with upregulation of p-NFKBIA and p-CHUK/IKBKB expression. NFKBIA prevents RELA from translocating to the nucleus, causing it to stay in an inactive state[25]. This is consistent with previous reports that SCFAs reduced macrophage induced inflammation by suppressing NF-κB signaling pathway and increasing IL10 expression[45]. IL10 inhibits NF-κB activity[26], which promotes inflammation, contributes to bone loss, and inhibits osteoblasts thereby decreasing bone formation[27]. This information suggests that inhibition of NF-κB signaling pathway may play a role in these processes by which VA inhibits osteoclast activity but enhances osteoblast activity. Although the in vitro study of RAW264.7 cells demonstrated a higher mRNA level of TNF (inflammatory cytokine) in the VA-treated group than the control group (Fig. 4e), we suspect it may be due to the basal level of TNF released by increased numbers of RAW264.7 cells[46]. VA inhibits differentiation of RAW264.7 cells into osteoclast-like cells, resulting in more RAW264.7 cells compared with the group treated with RANKL alone (Fig. 4a). Since this is the first study to report the protective effect of VA on postmenopausal BMD, additional questions about the underlying mechanisms by which VA impacts BMD provide new research opportunities with significant potential to impact the treatment of OP (e.g., the microbiota-SCFA-free fatty acid receptor [FFAR] signaling cascade may be a potential mechanism of VA because FFAR2 and FFAR3 are specific receptors of SCFAs[47]).

Since both *B. vulgatus* and VA were associated with BMD, we sought to determine whether *B. vulgatus* regulated bone metabolism by decreasing VA levels. Results from the association and MR analyses for *B. vulgatus*/VA-producing probiotics/VA in human data provide evidence to support a potential causal relationship between the relative abundance of *B. vulgatus* and serum VA levels. Although VA is important for health, the physiological concentrations are typically low in circulating blood[48]. Following disruption, GM of mice is known

to recover rapidly[49]. This may counter the inhibitory effect of *B. vulgatus* on VA, and account for the relatively slight decreases of VA we observed after *B. vulgatus* gavage. We have plans for future in vivo experiments with increased sample size and various doses/frequencies of *B. vulgatus* gavage that will have translational significance for potential clinical studies.

Supplementation with probiotics can benefit intestinal health and may be an effective therapeutic approach for preventing and/or treating bone loss[7,9,50,51] (e.g., *Lactobacillus reuteri* has been reported to reduce bone loss in elderly women with low BMD[9]). In the current study, we have observed negative correlations between *B. vulgatus* and VA-producing microbes (probiotics). VA improves BMD, and *B. vulgatus* appears to decrease BMD by suppressing growth of VA producing microbes, thereby depressing VA levels. Consequently, exploring the potential for VA-producing microbes (probiotics) to protect against OVX-induced bone loss can potentially be an innovative approach for OP intervention studies. Further mechanistic studies focused on *B. vulgatus*'s regulation of VA production by VA producing microges (e.g., the competitive and cooperative interactions between different GM species[52]) should be performed in the future. In addition, there may also be other potential pathways through which *B. vulgatus* regulates BMD that should be further explored. For example, *B. vulgatus* can alter bile acid metabolism and reduce IL22 secretion to cause increased disruption of ovarian function and insulin resistance for host health[34]. Bile acid metabolism, ovarian function, and insulin resistance are all associated with OP[53–55].

During data analysis process, to be confident that the findings are not likely spurious correlations, we calculated false discovery rates for multiple testing correction on most of the p-values. We did not perform false discovery rate for p-values from constrained linear regression analysis (Tables S1 and S5) since all bacteria species were tested in the same model. Although we may not have used the most stringent significance threshold for the identification of important bacteria species due to sample size limitations, we successfully validated the discovery results by subsequent in vivo and in vitro experiments, thus providing collectively solid results for the study as a whole.

There are several strengths of our study. First, to our knowledge, this is currently the largest shotgun metagenomics study directly testing associations between GM and human BMD. Second, we used stringent inclusion and exclusion criteria to ensure that subjects were relatively homogeneous for age, ovarian function and living environment, thereby minimizing potential confounding factors, and empirically enhancing the statistical power of our study. Third, functional mechanisms contributing to associations between GM and BMD were identified by using an innovative multi-omics approach to generate a comprehensive understanding of the crosstalk, interactions, and causal inference of the interactions between GM and human BMD, lending solid support with the big multi-omics data. Finally, we used both

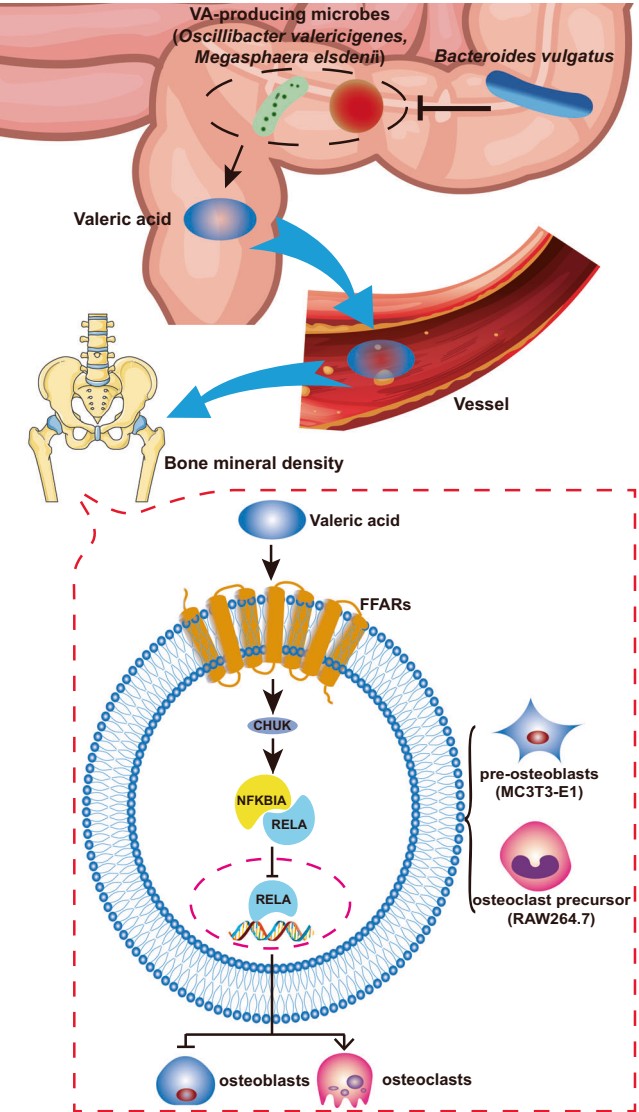

**Fig. 5 | Mechanisms of *B. vulgatus* on bone via VA production and NF-κB signaling pathway.** *B. vulgatus* inhibits valeric acid (VA)-producing species to reduce VA production within the gut. The VA suppresses pro-inflammatory RELA protein production in the osteoclast-like cells and osteoblasts to suppress maturation of osteoclast-like cells and promote maturation of osteoblasts. Thus, *B. vulgatus* decreases VA levels to enhance RELA protein production, which promotes inflammation and osteoclast activity, and suppresses osteoblast activity, then decreases bone mineral density (BMD).

statistical and experimental evidence to demonstrate the effects of specific bacterial species and GM-derived metabolites on BMD variation and regulation, and the causal effects of specific gut bacterial species on GM-derived metabolites in human serum. There are also some limitations that should be noted. For example, we did not have plasma SCFA profiling in the US white validation cohort.

In conclusion, we pioneered an innovative and comprehensive multi-omics approach for discovering associations between GM, SCFAs, and human BMD in a Chinese cohort, followed by in vitro and in vivo functional validation experiments as well as replication in an independent cohort of US white people. We successfully identified and validated individual gut bacterial species, and their derived metabolites, that were significantly associated with human BMD. We demonstrated that *B. vulgatus* appears to play a critical role in regulating bone metabolism through its effects on VA production. VA, in turn, inhibits bone resorption of OVX-mice, promotes

differentiation and mineralization of osteoblasts, and suppresses osteoclast differentiation and inflammatory responses. Our findings provide important insights into the pathophysiological mechanisms of OP from a perspective of human microbiota and their functional products, SCFAs, in human serum, and suggest potential novel bio-markers (e.g., VA) and treatment targets (e.g., elimination of *B. vulgatus* from the gut; dietary supplementation with VA) for OP prevention/intervention/treatment during the critical menopausal period as women age.

## Methods

This study was approved by the Third Affiliated Hospital of Southern Medical University (Guangzhou City, China) (ID: 201711012). It was performed under the principle of the Helsinki Declaration II. The release of the data by this work was approved by The Ministry of Science and Technology of the People's Republic of China (approval number 2023BAT0841 and 2023BAT1082). For ease of reference, we briefly mentioned the methods used in each of the respective results sections and provide a figure of the workflow (Fig. 6). Meanwhile, in order to increase readability, we spelled out full name of each abbreviation when it first appears in this manuscript, and we also provided a supplemental table for the summary of the abbreviations. Details are listed following and in the supplementary information.

### Subjects recruitment and sample preparation

517 independent, unrelated peri-/post-menopausal Chinese women were recruited. The inclusion criteria included (1) aged 40 years or older, (2) being in menopause stage, and (3) had lived in Guangzhou City for at least three months. Menopause is marked by the cessation of menstruation, where peri-menopause is a transition phase beginning at a woman's last menstrual cycle and continuing through the following 12 months without a menstrual cycle; once there are no menstrual cycles for 1 year, we term it post-menopause[56]. Briefly, exclusion criteria included the use of antibiotics, oestrogens, or anticonvulsant medications which may affect GM composition and/or bone metabolism in the past three months, as well as other diseases that could lead to secondary OP. A detailed list of exclusion criteria is shown in Table S7. We obtained signed informed consent from each subject before enrolling them into this study. Each subject filled out a questionnaire that collected information on age, medical history, family history, physical activity, alcohol consumption, diet habits, smoking history, nutrition supplements, etc.

We used a GE Lunar dual energy X-ray absorptiometry (DXA, GE Healthcare, Madison, WI, USA, version 13.31.016) with a standard scan model to measure the BMD of each subject at various skeletal sites, including the L1-L4, HTOT, FN, UD-RU. We performed DXA scanner calibration with a daily special phantom scan for quality assurance. The accuracy of BMD measurement was assessed by the coefficient of variation (CV%) of spine BMD, which was 0.89%.

Blood and stool samples were collected from each subject. Blood samples were collected after an overnight fast for >8 h and used for serum analysis and DNA extraction with the SolPure DNA Kit (Magen, Guangzhou, China). Each faecal sample was frozen at −80 °C within 30 min of sample procurement and used for GM DNA extraction with the E.Z.N.A.® Stool DNA Kit (Omega, Norcross, GA, USA). We stored the serum, blood, and GM DNA samples at −80 °C until further analyses.

### Metagenomic shotgun sequencing and annotation

Metagenomic shotgun sequencing and taxonomic/functional annotation of unigenes were performed by LC-Bio Technologies (Hangzhou) CO., LTD. ($n = 499$) (Hangzhou, China, www.lc-bio.com; this company owns an LC Sciences R&D department in Houston, TX, USA, www.lcsciences.com) with Hiseq 4000 (Illumina, San Diego, CA, USA), PE150 strategy and DIAMOND software.

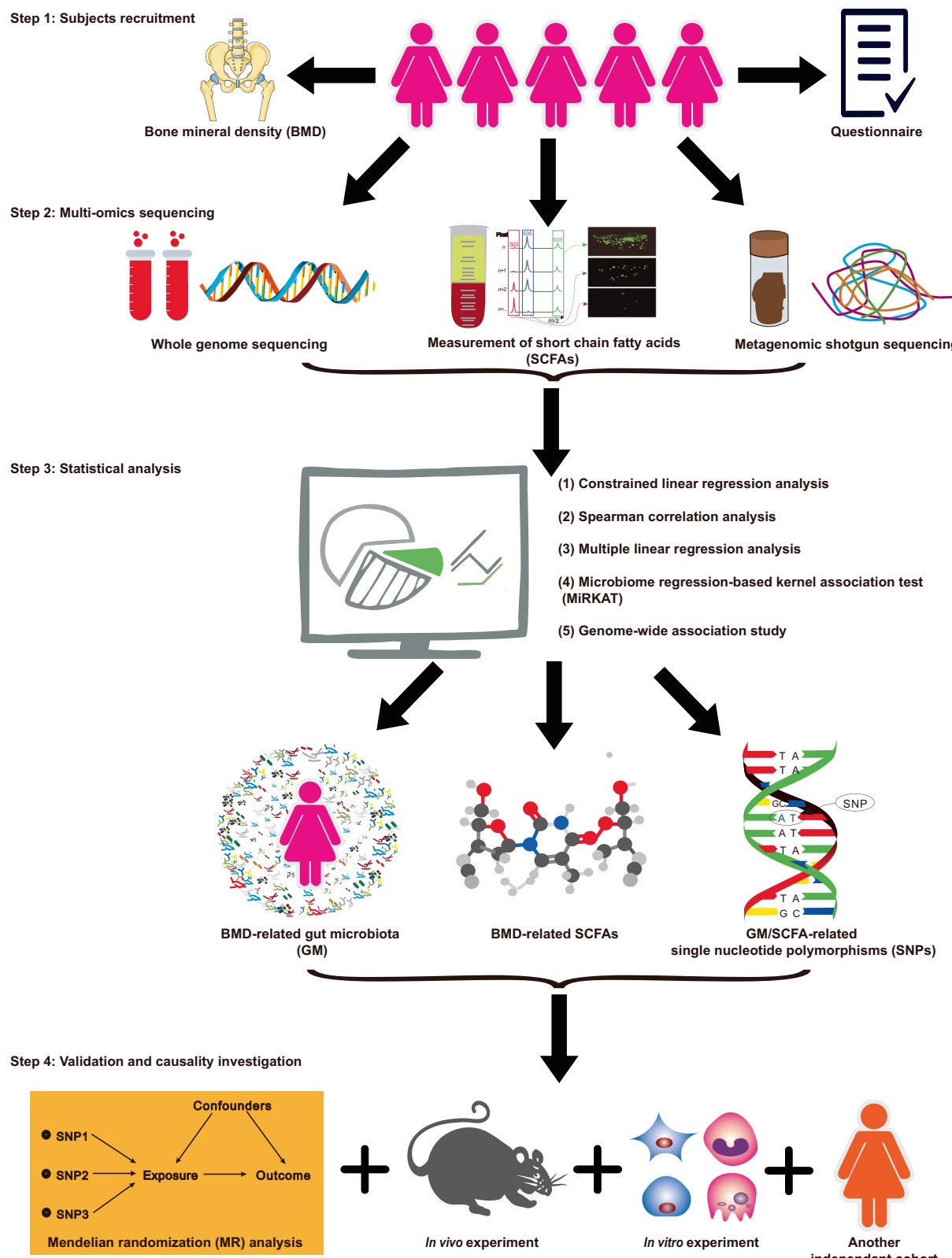

**Fig. 6 | Workflow of this study.** 517 peri-/post-menopausal Chinese women were randomly recruited from Guangzhou City in China. Their stool and blood samples were collected for metagenomics/targeted metabolomics/whole-genome sequencing. By performing various statistical association analyses for dual energy X-ray absorptiometry-derived BMD, several BMD-related bacterial species/SCFAs/GM functional capacity were identified. An independent cohort of US white people and MR analysis were used for validation/causality investigation. Finally, in vitro and in vivo functional experiments were performed to validate the findings.

Faecal DNA was fragmented by dsDNA Fragmentase (NEB, M0348S, Massachusetts, USA) and incubated at 37 °C for 30 min. Then, we used the fragmented cDNA to construct libraries. Blunt-end DNA fragmentation and size selection were performed with provided sample purification beads. An A-base was added to the blunt ends of each strand for the preparation of ligation to indexed adapters. These adapters also contained sequencing primer hybridization sites for single, paired-end, and indexed reads. The ligated products were amplified with polymerase chain reaction (PCR) under the following conditions: initial denaturation at 95 °C for 3 min, followed by 8 cycles of 98 °C for 15 s (denaturation), 60 °C for 15 s, 72 °C for 30 s (extension), then a final elongation at 72 °C for 5 min. Raw sequencing reads were processed in multiple cleaning steps. First, we removed sequencing adapters from sequencing reads by Cutadapt v1.9. Second, we trimmed low quality reads by Fqtrim v0.94. We used a sliding-window (size = 6 bp) to calculate the average quality of the bases within this window, and trimmed 3' end of reads when the average quality value dropped below 20. We also discarded the reads when the length was <100 bp and the percentage of "N" was larger than 5% after trimming. Third, we aligned reads to the host genome by using Bowtie2 v2.2.0, and removed host genomic contamination. Once quality-filtered reads were obtained, they were de novo assembled to construct metagenomes for each sample by SPAdes v3.10.0. The coding sequences (CDS) of metagenomic contigs were predicted by MetaGeneMark v3.26. The CDS of all samples were clustered by CD-HIT v4.6.1 to obtain unigenes. We used "blastp" function of DIAMOND v2.0.5 for unigene alignments. The unigenes were aligned against the NCBI NR database (ftp://ftp.ncbi.nlm.nih.gov/blast/db/FASTA/nr.gz) by DIAMOND software with the lowest common ancestor algorithm for taxonomic annotation, and against protein reference based on the KEGG pathway dataset (https://www.genome.jp/kegg/pathway.html) for functional annotation. It determines bit score and expected value (E-value) of the computed alignment. The bit score gives an indication of how good the alignment (hit) is; the higher the score, the better the alignment. The E-value gives an indication of the statistical significance of a given pairwise alignment; the lower the E-value, the more significant the hit. We selected the best hit with the highest bit score from the all potential hits (E-values $\leq 1 \times 10^{-5}$) as the respective KEGG Orthology (KO) for each unigene. KOs were further mapped to gut microbiota-associated functional KEGG modules. The relative abundance of each annotated functional module was the sum of relative abundance of the unigenes within this module.

The relative abundance of unigenes for a sample was estimated by transcripts per kilobase million (TPM, Formula 1, where k was the kth unigene, r was number of unigene reads, and L was unigene length) based on the number of aligned reads and the unigene length by Bowtie2 v2.2.0.

$$G_k = \frac{r_k}{L_k} \times \frac{1}{\sum_{i=1}^{n} \frac{r_i}{L_i}} \times 10^6 \tag{1}$$

### Measurement of SCFAs
Concentrations of SCFAs were measured with gas chromatography-tandem mass spectrometry (GC-MS/MS, 7890B-7000D, Agilent Technologies Inc, Santa Clara, CA, USA) (n = 500) by Wuhan Metware Biotechnology Co., Ltd (Wuhan City, China, www.metware.cn).

Serum samples were vortex-mixed with 36% phosphoric acid solution and further extracted supernatants (2 µl) by liquid-liquid extraction with methyl tert-butyl ether which containing SCFA stock solutions. Detailed conditions for GC-MS/MS based on a fused silica capillary column (DB-FFAP, 30 m × 0.25 mm × 0.25 µm, Agilent Technologies Inc, Santa Clara, CA, USA) were as follows: setting the injector temperature at 240 °C, keeping the initial oven temperature at 90 °C for 1 min, then gradually raising it to 140 °C, 160 °C, 200 °C, and 240 °C

at a rate of 10 °C/min, 5 °C/min, 15 °C/min, and 10 °C/min, respectively. Pure helium was used as a carrier gas at 1.0 ml/min rate. The main conditions of mass spectrometry included electron impact ion source and multi-reaction monitoring scan mode. The temperature of the transfer line, ion source, and quad were 240, 230, and 150 °C, respectively. The electron energy was 70 eV, and the solvent delayed 2.4 min. The process of QC and intra-day/inter-day accuracy were as follows: the QC samples were composed of SCFA stock solutions dissolved in methyl tert-butyl ether. Then the QC samples were processed in parallel with test samples to analyze detection stability and repeatability under the same process with an injection volume of 2 µl. Three QC samples were continuously injected to test the instrument stability. For every 10 test samples injected, a QC sample was inserted to check for the repeatability of the instrument. Intra-day and inter-day accuracy were evaluated and reported as CV% of repeatability at the concentration of each SCFA. The intra-day accuracy is 0.42–3.64%, and the inter-day accuracy is 1.12–3.40%, indicating a good stability of the instrument. We performed qualitative and quantitative analysis of SCFAs with Agilent Mass Hunter software. By using a stock solution containing mixtures of SCFAs, five calibration standards (concentration range from 0.005 to 8 mg/L) were prepared. Then we performed GC-MS/MS measurement as described above and integrated the obtained signals (e.g., retention time and peak area) to calculate relative retention time and area ratios. Meanwhile, the calibration curves were constructed by plotting the peak area versus concentration for each individual SCFA. And the slopes of the calibration curves were determined by performing linear regression analysis. In addition, average area ratio of blank samples was used as background signal/intercept. Finally, we calculated the concentrations of SCFAs with the area ratio, average area ratio blank samples, and slope.

### Association analysis among GM, SCFA, and BMD
To explore the association among GM, SCFA and BMD, in the Chinese cohort, we treated BMD as a continuous variable and then explored the associations among GM, SCFA, and BMD by using several association analysis methods. We first counted the number of gut species at different sample sizes. The R package "ggplot2" was used to generate a rarefaction curve. We plotted the number of species (Y-axis) against the number of samples (X-axis).

The α-diversity of GM was estimated based on the species profile of each sample according to the Shannon index by using R package "vegan". Then, partial Spearman correlation analysis was performed to estimate the association between the Shannon index and BMD variation. Using the program MiRKAT[13], we first calculated different kinds of kernels (including weighted UniFrac, unweighted UniFrac, and Bray-Curtis distance metrics), and then obtained the optimal kernel to estimate the association between GM species profiles and BMD variations. We also chose the modules whose relative abundance >0.10% to estimate the association between the KEGG modules and BMD variation by performing partial Spearman correlation analysis. In these analyses, we adjusted for a number of covariates, including age, BMI, YSM, FSH, exercise, and family annual income. Results with p-values <0.05 were considered statistically significant. False discovery rate was calculated for multiple testing correction on the p-values.

For the association analysis of species and BMD variation, we focused on non-rare species (relative abundance >0.10%) because rare species typically contribute significantly less to functional diversity than non-rare species due to their low relative abundances[57]. In addition, the rare species are generally difficult to cultivate, which might complicate the subsequent experimental validation in vivo. We first used the "igraph" package and the "psych" package of R software to identify strong correlations between the relative abundances of bacterial species based on the pairwise Spearman correlations. We also used the same method to identify correlations between specific bacterial species and VA-producing probiotic bacteria. For those highly

correlated bacterial species (with correlation coefficients [γ's] ≥ 0.80) that might cause a multiple co-linearity problem, we retained the bacterial species with a higher relative abundance and removed others as bacteria with relatively higher relative abundance may render higher power in association testing.

Constrained linear regression analysis[15] was then performed by using Stata 14 software with the "cnsreg" function, in which BMD variation was considered as the dependent variable, and the centered log ratio-transformed relative abundances of all bacterial species were set as the independent variables and analyzed simultaneously. The constraint ensures that the regression coefficients of all bacteria species sum to zero, an approach which is frequently used in compositional data analysis. The formula of constrained linear regression analysis is shown below (Formula 2), where p was the phylum level, $n$ was the number of phyla, s was species level, and m was the number of bacterial species in a given phylum. A number of covariates were adjusted in the regression analysis including age, BMI, YSM, FSH, exercise, and family annual income, to adjust and account for any of their potential effects on our analyses. We performed association analysis between the covariates and BMD/SCFAs at first, and only the significantly associated covariates with $p$-values of correlation coefficients <0.05 were included into the regression model. The threshold of exercise was 2.5 h per week based on a published guideline[58]. The family annual income was separated according to the local economic report. Bacterial species with a $p$-value of the regression coefficient (β) <0.05 were defined as BMD-associated bacterial species.

$$Y(BMD) = \sum_{p=1}^{n} \sum_{s=1}^{m} \beta_{ps} \mathrm{CLR}(X_{ps}) + \beta * \text{covariates}$$
$$\sum_{s=1}^{m} \beta_{ps} = 0 \tag{2}$$

Next, we used multiple linear regression analysis to identify BMD-associated SCFAs (Formula 3), where we performed log transformation for the SCFA concentrations due to their non-normal distributions and adjusted the same set of covariates mentioned above.

$$Y(BMD) = \sum \beta_i * \log(X_i) + \beta_j * \text{covariates}_j \tag{3}$$

Finally, we performed constrained linear regression analysis (Formula 4) to identify SCFA-associated bacterial species (non-rare species), while adjusting for age, BMI, YSM, and FSH as covariates.

$$Y(\log(\text{individual SCFA})) = \sum_{p=1}^{n} \sum_{s=1}^{m} \beta_{ps} \mathrm{CLR}(X_{ps}) + \beta * \text{covariates}$$
$$\sum_{s=1}^{m} \beta_{ps} = 0 \tag{4}$$

To replicate the association between GM and BMD, a total of 59 US white females, aged 60 years or older, were recruited from the ongoing Louisiana Osteoporosis Study (from 2011 to current), which aims to build a large sample pool in greater New Orleans and surrounding areas in Southern Louisiana, USA for investigating genetic and environmental factors for musculoskeletal disorders[59]. This study was approved by the institutional review board of Tulane University (New Orleans, LA, USA), and a written consent form was signed by each participant before data and bio-sample collection. Individuals who had pathological conditions that may influence BMD (e.g., a bilateral oophorectomy, chronic renal failure, liver failure, lung diseases, gastrointestinal diseases, and inherited bone diseases), or may influence GM (e.g., taking antibiotics, having gastroenteritis, major surgery involving hospitalization, and inter-continental travel in the past three months) were excluded. For each participant, BMD was measured by using a Hologic Discovery-A DXA machine (Hologic Inc., Bedford, MA, USA) and information on age, medical history, physical activity,

alcohol consumption, diet habits, smoking history, and nutrition supplements was assessed by a questionnaire. The accuracy of BMD measurement as assessed by the CV% of L1-L4 BMD was 0.54%. The metagenomic shotgun sequencing on the collected faecal samples was performed by Alkek Center for Metagenomics and Microbiome Research, Baylor College of Medicine. We used constrained linear regression analysis (Formula 2), as described above, to identify BMD-associated individual bacterial species. The covariates included age, BMI, exercise, and bone fracture.

To evaluate the overall effects of *B. vulgatus* on BMD, we further performed one-sided sign test[22] based on the coefficients of the regression of individual skeletal sites with *B. vulgatus*. The null hypothesis (H0) is that regression coefficients are equal to zero. The alternative hypothesis (H1) is that regression coefficients are less than zero.

### In vivo mouse experiment with *B. vulgatus*/valeric acid

All the procedures involving mice were approved by the Ethical Committee of Experimental Animal Science and Technology of Guangdong Medical Laboratory Animal Center (Foshan City, China). Seven-week-old female C57BL/6J mice (specific pathogen free [SPF] grade, $n = 108$) were purchased from the Guangdong Medical Laboratory Animal Center (Foshan City, China) and raised there. Five mice were kept in one box. The ambient temperature and humidity are 20–26 °C and 40%–70%, respectively. The dark/light cycle is 12 h: 12 h. The mice were fed freely with food and water for one week to acclimate to the new environment before experiments, and then were separated randomly into nine intended treatment groups ($n = 12$/group): Group I - blank, Group II - sham operated + NS gavage, Group III - sham operated + *B. vulgatus* gavage, Group IV - sham operated + normal drinking water, Group V - sham operated + drinking VA solution, Group VI - OVX + NS gavage, Group VII - OVX + *B. vulgatus* gavage, Group VIII - OVX + normal drinking water, Group IX - OVX + drinking VA solution.

OVX can result in a marked bone loss due to increased bone resorption with estrogen deficiency[60]. To evaluate potential causal effects of *B. vulgatus* on menopausal bone metabolism (focusing on lumbar vertebrae) and VA levels, we performed OVX on 8-week-old female C57BL/6J mice ($n = 12$/group) to model post-menopausal status. Bilateral OVX surgery was performed on the Group VI to IX of mice with general anaesthesia by the dorsal approach, which is a commonly used animal model for postmenopausal osteoporosis[60]. The same surgery process without resection of ovary (sham operated) was performed on the Group II to V of mice.

After one week for recovery, Group III (sham + *B. vulgatus*) and Group VII (OVX + *B. vulgatus*) mice were gavaged with 100 μl of *B. vulgatus* (ATCC8482, $2.5 \times 10^9$ colony-forming units/mouse, ATCC, Manassas, VA, USA; lot number: 70009621), while Group II (sham + NS) and Group VI (OVX + NS) mice were gavaged with 100 μl of NS. We referred to previous publications[61] about GM experiments for the dose of *B. vulgatus* for our study. The *B. vulgatus* was cultivated according to the product description and harvested in log phase for gavage. All mice were gavaged every other day and continued to feed freely with common food and water under the same environmental conditions. After gavaging for eight weeks, the mice were sacrificed and used for blood and bone measurements.

The Group V (sham operated + drinking VA solution) and Group IX (OVX + drinking VA solution) mice were fed with VA (1.5 mM), the Group IV (sham operated + normal drinking water) and Group VIII (OVX + normal drinking water) mice were fed with water without VA treatment (0 mM). We chose these doses based on our results of VA in vitro experiments (Fig. S3) and a consideration regarding the difference of VA concentration between extracellular fluid (~20% of body weight) of mice and cells in vitro. Therefore, we increased the VA concentration from 1.0 to 1.5 mM when it was administered into the mice. We did not use higher VA concentration since it can lead to a pH-

value lower than mouse stomach pH[62], and the purpose of our in vivo/in vitro experiments is to verify the protective effect of VA. We referenced a previous study[16] to allow the mice to drink water (with/without VA) freely. After eight weeks, the mice were sacrificed for bone/serum measurements.

The bones were prefixed in 4% paraformaldehyde (Solarbio, Beijing, China) at first. To evaluate bone histomorphology, we used µCT100 (Scanco Medical AG, Bassersdorf, Switzerland) with a resolution of 7.4 µm to measure the quantitative indices of the 5th lumbar vertebral body of the mice (e.g., Tb.N, Tb.Th, Tb.Sp and BV/TV) and to visualize the typical bone microstructure changes for illustration.

Half of the mice bone samples ($n = 6$/group) were incubated with 10% ethylene diamine tetraacetic acid (EDTA, pH = 7.4, Solarbio, Beijing, China) until the bone was easily penetrated through by a needle without any force. Subsequently, samples were dehydrated in graded ethanol solutions and infiltrated and embedded in paraffin. Then, for each specimen, 5 serial sections (5 µm thick) were cut on a microtome to perform immunohistochemistry-osteocalcin staining (Abcam, Cambridge, UK, 1:1000) and tartrate-resistant acid phosphatase (TRAP) staining (Wako, Osaka, Japan) to respectively assess osteoblastogenesis and osteoclastogenesis of trabecular bone in the 5th lumbar vertebrae. Cells per bone perimeter was used to calculate the number of positive cells. The other mice bone samples ($n = 6$/group) were infiltrated and embedded in resin. The resin block was sectioned using a microtome and a 6 µm thickness section was prepared to perform hematoxylin-eosin staining (ZSGB-BIO, Beijing, China) and Von Kossa staining (Servicebio, Wuhan, China) to assess bone trabecula and bone mineralization, respectively. We also used enzyme-linked immunosorbent assay (ELISA) kits to measure serum levels of PINP and CTX-I (Elabscience, Wuhan, China), which are markers of bone formation and bone resorption, respectively.

For the statistical analysis, as in previous study[61], non-parametric one-way analysis of variance (ANOVA, Kruskal-Wallis test) was performed to identify the differences of relative abundance of B. vulgatus and VA-producing bacteria (Megasphaera elsdenii and Oscillibacter valericigenes)/VA levels/quantitative indices of trabecular bone volume and structure among different mice groups. We further used the non-parametric (Mann–Whitney test)/t-test to identify the difference between groups if the ANOVA results are significant. False discovery rate (q-value) was calculated for multiple testing correction on the p-values (two-sided).

## MR analysis for BMD-associated GM and SCFA

Whole-genome sequencing was performed by BGI Genomics Co. Ltd (Shenzhen, China; https://www.genomics.cn/) with BGISEQ-500 platform for the same Chinese cohort ($n = 500$). Qualified genomic DNA samples were randomly fragmented by Covaris technology and the fragment of 350 bp was obtained after fragment selection. End repair of DNA fragments was performed and an "A" base was added at the 3′-end of each strand. Adapters were then ligated to both ends of the DNA fragments, and ligation-mediated PCR was performed for amplification, single strand separation and cyclization. The rolling circle amplification was performed to produce DNA Nanoballs. The qualified pair-end reads were read through on the BGISEQ-500 platform. High throughput sequencing was performed for each library to ensure that each sample meets the average sequencing coverage requirement. Sequencing-derived raw image files were processed by BGISEQ-500 Base-calling software for base-calling with default parameters. Sequence data of each individual were generated as paired-end reads, which is defined as "raw data" and stored in FASTQ format. Clean reads were produced by data filtering, including removed reads containing sequencing adapter, removed reads whose low-quality base ratio (base quality less than or equal to 5) was more than 50%, removed reads whose unknown base ("N" base) ratio was more than 10%. Then, clean

reads of each sample were mapped to the human reference genome (GRCh38/HG38). Burrows-Wheeler Aligner v0.7.17 software was used for sequence alignment. To ensure accurate variant calling, we followed recommended best practices for variant analysis with the Genome Analysis Toolkit 4.

After obtaining WGS data, we used PLINK 1.9 software with the default criteria as follows for SNPs quality control: the SNPs with missing rate <0.1, minor allele frequencies > 0.01, and Hardy-Weinberg equilibrium p-value $> 1.0 \times 10^{-5}$ were included for GWAS analysis. A QQ plot of the association statistics was used to evaluate the effects of potential confounding due to population structure. We then performed the GWAS to identify B. vulgatus and VA-associated SNPs.

Since SCFAs are produced exclusively by the GM[16], we investigated the causality of B. vulgatus exposure on VA outcome with one-sample MR analysis (rather than bidirectional MR analysis), which can provide unbiased estimates of the causal effect without the need for a traditional randomized clinical trial by using SNPs as IVs[21]. For the implementation of MR, we first selected independent genetic variants ($r^2 \leq 0.01$) associated with B. vulgatus (p-values $< 1 \times 10^{-5}$) as the IVs. We then obtained the corresponding effect estimates of these IV SNPs from the VA GWAS analysis. To ensure the SNPs used as IVs for B. vulgatus are not in LD with one another, a vital assumption of MR, we calculated pairwise-LD between all of our selected SNPs in the 1000 Genomes European reference sample using PLINK 1.9. For all the pairs of SNPs determined to violate the independence assumption with $r^2 > 0.01$, we retained only the SNPs with the smaller B. vulgatus association p-values. The MR causal effect estimate for each instrument was calculated as the ratio of the VA and B. vulgatus GWAS estimates. The causal effects from multiple instruments were combined using various meta-analysis approaches including weighted median method, maximum likelihood estimation and inverse-variance weighting (IVW). In order to assess the presence of horizontal pleiotropy, a critical assumption of MR, we tested the significance of the MR-Egger regression intercept.

## Effects of VA on osteoclastogenesis, and osteoblastogenesis

To observe the differentiation of osteoclast-like cells, we used the murine monocyte/macrophage cell line RAW264.7 (Beina Chuanglian Biotechnology Institute, Beijing, China) as osteoclast precursors, which can differentiate into osteoclast-like cells in the presence of RANKL (50 ng/mL, PEPROTECH, Rocky Hill, NJ, USA). The cells were grown in alpha-minimum essential medium (αMEM, GIBCO/Invitrogen, Carlsbad, CA, USA), containing 10% fetal bovine serum (FBS, GIBCO/Invitrogen, Carlsbad, CA, USA), 100 U/ml penicillin and 100 mg/ml streptomycin sulfate (GIBCO/Invitrogen, Carlsbad, CA, USA), at 37 °C in a humidified atmosphere of 95% air and 5% $CO_2$. The cells in 12-well plate were used for staining, in six-well plate for RNA/protein extraction. We performed in vitro experiments with various concentrations of VA (TCI, Tokyo, Japan) (0, 0.01, 0.1, 1 mM) to assess suitable concentration for cell differentiation. We changed the medium every other day and fixed the cells at the end of the 5th day. We then performed TRAP staining (Wako, Osaka, Japan) to observe TRAP-positive cells (TRAP⁺MNCs). The number of TRAP⁺MNCs/well were counted under a light microscope for a two-sample t-test.

To observe the differentiation of osteoblasts, MC3T3-E1 cells (Beina Chuanglian Biotechnology Institute, Beijing, China) were maintained in osteoblastogenic medium (Oricell, Guangzhou, China) with 10% FBS, 100 U/ml penicillin, and 100 mg/ml streptomycin sulfate at 37 °C with 5% $CO_2$ and treated with various concentrations of VA (0, 0.01, 0.1, 1 mM) for 14 days. ALP staining (Beyotime, Shanghai, China) and ARS staining (Oricell, Guangzhou, China) were performed according to standard techniques to evaluate osteoblast differentiation and extracellular matrix mineralization, respectively. Based on previous studies[63,64], we used a commercialized ALP activity

colorimetric assay kit (KeyGEN, Nanjing, China) with a measurement of solution absorbance, to obtain quantitative data for ALP; evaluated extracellular matrix mineralization via quantifying areas of mineralized nodules stained with ARS.

## Gene expression in osteoclast-like cells and osteoblasts

Total protein and RNA were extracted from the osteoclast-like cells and osteoblasts to examine expression of the selected genes. Since the NF-κB signaling pathway has been reported to activate osteoclastogenesis and inhibit osteoblastogenesis[27], and SCFAs can suppress this pathway[45], we explored whether the suppressive effect and stimulating effect of VA on osteoclast-like cells and osteoblasts, respectively, were at least partially explained by inhibition of the NF-κB signaling pathway. Western blot analysis was performed with NF-κB signaling pathway-associated primary antibody targeting p-RELA (Absin, Shanghai, China, dilution 1:2000), p-CHUK, p-IKBKB and p-NFKBIA (Cell Signaling Technology, Danvers, MA, USA, dilution 1:1000). Real-time quantitative polymerase chain reaction (qPCR) analysis (SYBR green real time PCR master mix, TOYOBO, Osaka, Japan) was performed for the transcriptional levels of TNF (sense: GGTGCCTATGTCTCAGCCTCTT; antisense: GTGGTTTGTGAGTGTGAGGGTCT) and IL10 (sense: CGGGAAGACAA-TAACTGCACCC; antisense: GTCGGTTAGCAGTATGTTGTCCAG), calculated by using the comparative threshold cycle method[65]. GAPDH (Cell Signaling Technology, Danvers, MA, USA, dilution 1:1000) and mouse-18S-rRNA (sense: AGTCCCTGCCCTTTGTACACA; antisense: CGATCCGAGGGCCTCACTA) were used as internal references for western blot and qPCR respectively. We examined IL10 because it can be increased by SCFAs[45] and inhibit activation of NF-κB[26].

## Reporting summary

Further information on research design is available in the Nature Portfolio Reporting Summary linked to this article.

## Data availability

The data that support the findings of this study have been deposited in public databases. The sequencing data of the WGS can be found in "Genome Sequence Archive for Human" (https://ngdc.cncb.ac.cn/gsa-human, accession No. HRA004900). The metagenomic sequencing data can be found in "Sequence Read Archive" (https://www.ncbi.nlm.nih.gov/sra, accession No. PRJNA986283 and PRJNA1011937). The GWAS data, characteristics of the cohorts, serum SCFA levels, and relative abundance of GM and KEGG modules have been deposited in the Figshare database (https://figshare.com, https://doi.org/10.6084/m9.figshare.23267351). All the data are subject to open access. Source data are provided with this paper.

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

## Acknowledgements

HW Deng and H Shen were partially supported by grants from the National Institutes of Health [U19AG05537301, R01AR069055, P20GM109036, R01MH104680, R01AG061917], and the Edward G. Schlieder Endowment and the Drs. W. C. Tsai and P. T. Kung Professorship from Tulane University. J.S. was partially supported by grants from the Science and Technology Program of Guangzhou, China [201604020007], and the National Natural Science Foundation of China [81770878]. H.M.X. was partially supported by the National Key R&D Program of China [2016YFC1201805 and 2017YFC1001100]. X.L. was partially supported by the Scientific Research Start Plan of Shunde Hospital of Southern Medical University [SRSP2021007] and the Foshan and Shunde postdoctoral research funding.

## Author contributions

H.W.D. conceived, designed, initiated, and directed the whole project, revised, rewrote/re-structured some sections and finalized the manuscript. J.S. and H.M.X. managed the study done in their institutions. X.L. performed the data analysis, drafted the manuscript and conducted in vivo/in vitro experiments. F.Y.L. and C.L.G. contributed

to the data analysis. R.G., D.Y.P., Z.C., and Z.F.L. performed clinical diagnosis and recruited subjects. C.P., X.J.X., Y.C.C., R.Z., X.F.W., Z.X.A., J.M.L., Y.Q.S., and Y.H.Z. collected samples and clinical phenotypes. N.J.Y., Q.Z., B.Y.G., H.M.L., W.Q.L., Z.L., C.J.P., X.M.S., J.G., Q.Z., K.J.S., X.H.M., and H.S. contributed to text revision and/or discussion.

## Competing interests

The authors declare no competing interests.

## Additional information

[1]Shunde Hospital of Southern Medical University (The First People's Hospital of Shunde), No.1 of Jiazi Road, Lunjiao, Shunde District, Foshan City 528308 Guangdong Province, China. [2]Department of Endocrinology and Metabolism, The Third Affiliated Hospital of Southern Medical University, Guangzhou 510630 Guangdong Province, China. [3]Center of System Biology, Data Information and Reproductive Health, School of Basic Medical Science, Central South University, Changsha 410008 Hunan Province, China. [4]Tulane Center for Biomedical Informatics and Genomics, Deming Department of Medicine, School of Medicine, Tulane University, New Orleans, LA 70112, USA. [5]LC-Bio Technologies (Hangzhou) CO., LTD., Hangzhou 310018 Zhejiang Province, China. [6]Departments of Neuroscience & Regenerative Medicine and Orthopaedic Surgery, Medical College of Georgia, Augusta University, Augusta, GA 30914, USA. [7]Department of Preventive Medicine, College of Medicine, University of Tennessee Health Science Center, Memphis, TN 38163, USA. [8]Department of Biostatistics, University of Alabama at Birmingham, Alabama 35294, USA. [9]Department of Biomedical Sciences, School of Medicine, University of Missouri-Kansas City, 2411 Holmes Street, Kansas City, MO 64108, USA. [10]These authors contributed equally: Xu Lin, Hong-Mei Xiao. ✉e-mail: hmxiao@csu.edu.cn; shenjiedr@163.com; hdeng2@tulane.edu

