## [Peer Review File · Nature Communications]

Gut microbiota impacts bone via *Bacteroides vulgatus*-valeric acid-related pathwaysReviewers' comments:

Reviewer #1 (Remarks to the Author):

- 1) It was not clear how false discovery rate is incorporated in the interpretation of the results, as only p-values are discussed in the written results.
- 2) Line 127. It is stated that although no significant association was found between GM biodiversity and BMD at L1-L4 or HTOT, we hypothesized that individual bacterial species contributed to these phenotypes. Why? Spine and hip are critical sites. Why no correlations? How do the Authors explain difference between sites?
- 3) The authors found that several bacterial species correlated with bone density at one site but not another. This site specificity is not explained. How can a bacterial species influence BMD at one site but not another?
- 4) Authors focused on Valeric acid, which is a minor SCFA. What about more abundant SCFAs, such as butyrate and propionate? Were these metabolites analyzed? Did they correlate with BMD at any site? How was the identity of valeric acid confirmed? Moreover, it is not clear why plasma SCFA were not also tested in the validation cohort of Caucasians.
- 5) Table 2: Confidence in the findings and interpretations would be higher if SCFA production or metabolism had emerged as a functional capacity. Also, pathways related to glucose metabolism seemed to dominate in this table, but this finding is largely ignored.
- 6) *B. vulgatus* was negatively associated with L1-L4 in the Chinese population while it was negatively associated with HTOT in the US population. Why *B. vulgatus* was associated with one site and not the others? And why *B. vulgatus* was associated with different sites in the two populations?
- 6) The animal studies are insufficient. No sham operated control is provided. Therefore, there is no evidence that ovx induced bone loss. The effect of *B. vulgatus* in control animals was not assessed. *B. vulgatus* could have distinct effects in eugonadic mice and estrogen deficient mice. Quantitative indices of trabecular (and cortical) bone volume and structure should be provided, including BV/TV, Tb.Th, Tb.N and Tb.Sp. The qualitative data shown in figures 2d and 2e are not acceptable. Assessment of bone turnover by quantitative histomorphometry would have been more robust.
- 7) Figure 3a and 3b OC and osteoblast number should be counted.
- 8) IL-10 is a cytokine produced by regulatory T cells. The meaning of the observed increase in IL-10 is unclear.

Reviewer #2 (Remarks to the Author):

Xu Lin and colleagues have investigated the possible role of the gastrointestinal microbiome composition in Bone Mineral Density. They associate the abundance of a single species *B. vulgatus* with BMD in their cohort and replicate this finding in an independent dataset. Furthermore, using multiple -omics in vitro and in vivo experiments they demonstrate the possible causality of *B. vulgatus* on BMD. The research and work presented is extensive and has great value for the field of BMD research, as the microbiome presents a modifiable risk factor which could be a novel therapeutic target for BMD (osteoporosis). The authors have performed a tremendous amount of work and follow-up experiments to demonstrate the role of *B. vulgatus* in BMD. However, the statistical evidence of association for *B. vulgatus* is very

weak, casting serious doubt on the interpretation and conclusions drawn by Xu Lin and colleagues. This is mainly due to the (wrong) choice of statistical methodology.

Major comments:

1. Consistently throughout the manuscript no multiple testing correction (Bonferroni, False Discovery Rate) is applied on p-values. How are the authors certain that their results are not spurious correlations? (Especially for the single microbiome species analysis this should have been done as still ~62 individual species were examined).
 - a. In the KEGG modules and BMD analysis results were significant if $p < 0.01$, why is the P-value threshold lowered here, and not in other analysis? Why not multiple testing correction in this analysis?
 2. The authors consistently use statistical methods not suited for the analysis of compositional data such as metagenomics data. These traditional (non-compositional) methods can appear to give satisfactory results. However, these results can be misleading and unpredictable, thus I strongly recommend the authors to perform their association analysis using compositional methods(1).
 - a. This includes the Bray-Curtis dissimilarity matrix: the use of Bray-Curtis with compositional data (microbiome data) is very problematic, especially if the data has not been normalized (on read count) (1). (no mention of normalization procedures were found in the manuscript or extended data, nor average read count/range of read counts or if samples with low read count were excluded). The authors should have used methods which take the compositional nature of the microbiome data into account: log-ratio normalization then Aitchison distance matrix followed by the association analysis(MirKAT/PerANOVA ect.)
 - b. Also for the single microbiome species analysis methods which take the compositional nature of the microbiome data into account should have been used examples: MaAsLin (takes correlation between species into account) or isometric log-ratio transformation (to remove potential collinearity in the data)(1,2,3).
 - c. The same comment is also valid for the use Spearman correlation Coefficient, there are compositional methods available(1).
 3. Could the authors describe which "common probiotic bacteria" they have used and pooled for the correlation analysis with *B. vulgatus* (in the text or in the supplementary methods)? Could they also present a table/data set which contains the aver. Abundancies of these "common probiotica bacteria"?
 4. For the MR analysis a GWAS was performed to discover SNPs associated with *B. vulgatus* and valeric acid, however currently I am not able to asses the results of this analysis as essential information regarding this analysis is not available or not (clearly) presented in the text:
 - a. What the sample size was for this GWAS, (how many individuals were whole genome sequenced and used for GWAS)
 - b. On which individuals was this GWAS performed, was this the same set as of which metagenomics and phenotype data was available? (if yes this might cause for bias issues in the MR analysis) or did you use the published data of LLDeep (ref 43 in manuscript?)
 - c. Were the selected SNPs both associated with *B. vulgatus* and valeric acid or one of the two?
 - d. Which SNPS and how many SNPs were used in the MR analysis?
 - e. Where are the summary statistics of the performed GWAS studies?
 5. *B. vulgatus* was only nominal associated with L1-L4-BMD (lumbar spine BMD, ED table 3). However, L1-L4 BMD was not correlated to BMD variation in Shannon index analysis and Kernal analysis (ED table 2). In addition, in the replication *B. vulgatus* was only nominal significant replicated in HTOT BMD (left total hip BMD). There is no consistent association of the microbiome or *B. vulgatus* with a single BMD site. Could the authors explain this discrepancy?
 6. The authors are commended for taking the effort to replicate their main findings in an independent cohort, however this cohort has a very small sample size of 59. Perhaps the authors could consider collaborating with large microbiome cohorts and consortia for more robust replication? (MiBioGen, LLDEEP, HeLIOS, Rotterdam Study, TwinsUK etc) whom all have microbiomics and phenotypic data (some also have metagenomics and metabolomics)
- Considering the above comments, it is my opinion that the current conclusions are not supported by the results, specifically the association of *B. vulgatus* with BMD, resulting in a manuscript unsuited for publication. However, it is also my opinion that if the authors address the above comments and using

compositional methods find an association with *B. vulgatus* their manuscript can be an exceptionally good one.

Minor Comments

- The term "gut microbiome" although commonly used is actually not the correct term. It only refers to the microbial composition in the "gut", while the stool microbiome is actually a representative of the microbiome composition of the entire oral-gastrointestinal microbiome. Thus your gut microbiome composition here actually is: the relative abundance of faecal bacteriome. Please consider this in your manuscript, by for example defining this in the introduction of the methods.
- Please give sample sizes in analysis and figures examples: how many non-rare species were there (abundance >0.10%)
- Figure 1: a nice figure but very difficult to read (especially for those who are colour blind like myself: Tritanomaly). Please consider only showing the distribution of genus level or different methods of representing all found species.
- The results of the single species association are now presented in the text, please consider to present these results in a table to increase readability (if journal constraints allow this).
- In the discussion the authors list the strength of their study, but do not mention any possible weaknesses or pitfalls of their own study. I find it good practice to also acknowledge own weaknesses or future improvement points of a study if strengths are mentioned.
- Please refrain from using such sentences: (discussion) we pioneered an innovative and comprehensive multi-omics approach for discovery in a Chinese cohort. Although you have done extensive research none of the used methods were novel or not published before.
- The discussion is quite lengthy and contains many repetitions, please consider restructuring the discussion to increase readability and to remove the unnecessary repetition.

References

- 1 Gregory B. Gloor, Jean M. Macklaim, Vera Pawlowsky-Glahn and Juan J. Egozcue, Microbiome Datasets Are Compositional: And This Is Not Optional. *Front. Microbiol.*, 15 November 2017
- 2 MaAsLin: <https://huttenhower.sph.harvard.edu/maaslin2/>
- 3 Aitchison, J., 1986, *The statistical analysis of compositional data: Monographs on statistics and applied probability*: Chapman & Hall Ltd., London, 416p.

Reviewer #3 (Remarks to the Author):

This is a fairly straightforward study that is clearly written and follows a logical flow. The analysis shows good bioinformatic and statistical knowhow. There are some concerns, however, that should be addressed.

1. As far as I can tell, while the study population seems substantial at first (517), only ~37 (7%) had osteoporosis. This is a small and probably underpowered number for metagenomic studies.
2. Related to this, the data should be stratified for osteopenia and osteoporosis.
3. In figure 1, it would help if the proportion of subjects with normal, osteopenia and osteoporosis were in some way projected onto the figure or shown separately/individually.
4. It is not clear how the variables such as age and BMI vary with BMD. This should be represented or clarified in the text.
5. Figure 2: this should preferably be dot plot and not simply bars
6. The authors also need to discuss the size of the effect *B. vulgatus* has on the BMD in the observational study.

Minor points:

GM is a non-conventional abbreviation. It is unnecessary, will slow down readers and irritate many of

them. It should be spelled out fully

As far as I can tell, most if not all of the relevant literature is cited. However, for one of the earlier studies, Das et al, it is mentioned (line 255) that a trend of an association between shannon diversity and BMD was shown. As I recall, the P-value in that report was above 0.1 and so cannot be called a trend.

[redacted]

We thank you for giving us the opportunity to appeal for our previous manuscript (ID: NCOMMS-20-12271A-Z). We have submitted our updated manuscript (ID: NCOMMS-22-17443) for consideration of publication in *Nature Communications* again. We also appreciate the reviewers for their valuable comments and helpful suggestions! Our responses to each of the reviewer's comments are summarized in the following.

Reviewers' comments:

Reviewer #1 (Remarks to the Author):

1) It was not clear how false discovery rate is incorporated in the interpretation of the results, as only p -values are discussed in the written results.

Response

We thank the reviewer for this comment. During the data analysis process, we calculated false discovery rate for multiple testing correction on most of the p -values to be confident that the findings are not likely spurious correlations (Table 2 and 3). We did not perform false discovery rate for p -values from constrained linear regression analysis (Table S1 and S5) since all bacteria species were tested in the same model.

Although we may not have used the most stringent significance threshold for the identification of important bacteria species due to the sample size limitation, we successfully validated the discovery results by *in vivo* and *in vitro* experiments, thus providing collectively solid results for the study as a whole. We mentioned these contents in the updated manuscript (Page 20 Line 18 – Page 21 Line 3).

2) Line 127. It is stated that although no significant association was found between GM biodiversity and BMD at L1-L4 or HTOT, we hypothesized that individual bacterial species contributed to these phenotypes. Why? Spine and hip are critical sites. Why no correlations? How do the Authors explain difference between sites?

Response

We thank the reviewer for pointing this out. The GM biodiversity was calculated using counts of individual species¹, so it is plausible that even if there are different individual bacterial species associated with BMD, the combinations of different species may lead to the same biodiversity, and which may be not associated with BMD (Page 6 Lines 14–17).

Meanwhile, it is not unusual and actually commonly observed in epidemiology studies that the association results for BMD may be different for different skeletal sites. This is particularly true and widely observed for genetic epidemiology studies². A prominent example is the association of the *WNT16* gene (a most robust associated gene with BMD identified in the field), which was associated with hip BMD in Caucasian and Asian populations in some studies³, but associated with lumbar spine BMD in European, Hispanic-American, and African-American in other studies⁴, while associated with forearm BMD only in European in other studies⁵. It is well accepted that BMDs of different sites are correlated and can together or individually reflect the bone health in general⁶. We mentioned these contents in the updated manuscript (Page 15, Line

21–Page 16, Line 8).

3) The authors found that several bacterial species correlated with bone density at one site but not another. This site specificity is not explained. How can a bacterial species influence BMD at one site but not another?

Response

We thank the reviewer for pointing this out. Take the *B.vulgatus* for example, we did find a negative association between *B. vulgatus* and spinal BMD ($\beta = -0.027$, p -value = 0.032) in the Chinese cohort (Figure 2c and Table S1), which was consistent with total hip BMD in the US cohort ($\beta = -0.018$, p -value = 0.029, Figure 2d and Table S5). Actually, the *B.vulgatus* was negatively associated with different sites of BMDs in both of the two cohorts (Table S6, some of the regression coefficients were non-significant may due to sample size limitation). Meanwhile, we performed a formal one-sided sign test⁷ based on the regression coefficients (Table S6). The null hypothesis (H0) is that regression coefficients are equal to zero. The alternative hypothesis (H1) is that regression coefficients are less than zero. The results indicated that there were negative associations between BMDs and *B.vulgatus* in both of Chinese and the US cohorts (Chinese p -value = 4.8×10^{-7} , Caucasian p -value = 9.8×10^{-4}) (Page 10, Lines 4–9).

As mentioned in the last response, it would not be an unusual phenomenon that the *B. vulgatus* was associated with different sites of BMDs in different studies because

genetic components play important roles in BMDs at different sites. Furthermore, the association between *B. vulgatus* and BMD we found in Chinese cohort was only for a discovery. We subsequently validated it by several methods, including another independent cohort and *in vivo* experiment. All of the results together provide convincing evidence for the association between *B. vulgatus* and BMD.

4) Authors focused on Valeric acid, which is a minor SCFA. What about more abundant SCFAs, such as butyrate and propionate? Where these metabolites analyzed? Did they correlate with BMD at any site? How was the identity of valeric acid confirmed? Moreover, it is not clear why plasma SCFA were not also tested in the validation cohort of Caucasians.

Response

We appreciate the reviewer for pointing out the confusion. We totally measured six kinds of SCFAs, including caproic acid, isovaleric acid, butyric acid, acetic acid, isobutyric acid, and valeric acid. All the SCFAs were measured by Wuhan Metware Biotechnology Co., Ltd (Wuhan City, China, www.metware.cn) with targeted metabolomics. During the measurement process, we used the authentic standards to confirm the chemical structures for the metabolite identification.

In our results, only valeric acid was correlated with BMD. Considering the limitation of words, we did not report other SCFAs in this manuscript, but we added them into the supplementary appendix (Table S2). We did not have plasma SCFA profiling in the

Caucasian validation cohort, and we added this limitation in our updated manuscript (Page 21, Lines 18–19).

5) Table 2: Confidence in the findings and interpretations would be higher if SCFA production or metabolism had emerged as a functional capacity. Also, pathways related to glucose metabolism seemed to dominate in this table, but this finding is largely ignored.

Response

We thank the reviewer for this comment. All the BMD-related functional capacities were listed in the Table 3; however, we failed to identify which was associated with SCFA production or metabolism. The glucose metabolism-related pathways (Table 3) were identified by genes of gut microbiota, which reflect function of gut microbiota rather than human. In our opinion, these pathways play important roles in bacterial cellular energy and growth. We have discussed these functions in the updated manuscript (Page 16 Line 18–Page 17 Line 5).

6) *B. vulgatus* was negatively associated with L1-L4 in the Chinese population while it was negatively associated with HTOT in the US population. Why *B. vulgatus* was associated with one site and not the others? And why *B. vulgatus* was associated was associated with different sites in the two populations?

Response

Please refer to our responses to the second and the third questions. Thank you!

7) The animal studies are insufficient. No sham operated control is provided. Therefore, there is no evidence that OVX induced bone loss. The effect of *B. vulgatus* in control animals was not assessed. *B. vulgatus* could have distinct effects in eugonadic mice and estrogen deficient mice. Quantitative indices of trabecular (and cortical) bone volume and structure should be provided, including BV/TV, Tb.Th, Tb.N and Tb.Sp. The qualitative data shown in figures 2d and 2e are not acceptable. Assessment of bone turnover by quantitative histomorphometry would have been more robust.

Response

We thank the reviewer for pointing out this issue. First, we purchased a group of normal (not OVX and not be treated with oral gavage) 18-week-old female C57BL/6J mice (n = 12) from the Laboratory Animal Center of the Southern Medical University (Guangzhou, China) to compare with the Group I (*B. vulgatus* gavage) and Group II (NS gavage) OVX-mice. Since all the mice in each group were purchased from the Laboratory Animal Center of the Southern Medical University (Guangzhou, China) and were raised in the same specific pathogen free environment in the same lab, it would be reasonable to compare these different groups of mice (Page 31 Lines 1–8) regarding levels of *B. vulgatus*, VA and BMD (workflow of the *in vivo* experiment is shown as follow). We observed an increased relative abundance of *B. vulgatus* in the OVX-mice as compared to the normal mice, and the OVX-mice with *B. vulgatus* gavage had the

highest relative abundance of *B. vulgatus* (p -value < 0.01, Figure 2e) (Page 11, Lines 4–6).

Figure: workflow of the *in vivo* experiment

Second, in the updated manuscript, we randomly selected eight mouse bone samples (the 5th lumbar vertebrae) from each of the two mice groups (Group I and Group II), and used µCT100 (Scanco Medical AG, Bassersdorf, Switzerland) to obtain quantitative indices (e.g., trabecular number [Tb.N], trabecular thickness [Tb.Th], and

trabecular separation [Tb.Sp]) and to visualize the typical bone microstructure changes between the two groups for illustration (Figures 2h and 3c; Page 31, Lines 14–19).

Third, we did not perform dynamic histomorphometry for quantitative data of bone turnover because it requires tetracycline before sacrificing the mice, which might affect the GM composition⁸. We added this limitation in our updated manuscript (Page 21 Line 21–Page 22 Line 1).

7) Figure 3a and 3b OC and osteoblast number should be counted.

Response

We thank the reviewer for this suggestion. The number of TRAP-positive cells (TRAP⁺MNCs)/well were counted under a light microscope for a two-sample t-test (Figure 4a; Page 35, Lines 5–7). We also referred to previous studies^{9,10}, used a commercialized alkaline phosphatase (ALP) activity colorimetric assay kit (KeyGEN, Nanjing, China) with a measurement of solution absorbance, to obtain quantitative data for ALP; evaluated extracellular matrix mineralization via quantifying areas of mineralized nodules stained with ARS (Figure 4b; Page 35, Lines 17–20).

8) IL-10 is a cytokine produced by regulatory T cells. The meaning of the observed increase in IL-10 is unclear.

Response

We thank the reviewer for this comment. In this study, we revealed that the protective

effect of VA is partially attributable to decreased osteoclast activity and increased osteoblast activity mediated by inhibition of NF- κ B p65 expression. Since IL10 can be increased by SCFAs¹¹ and inhibits activation of NF- κ B¹², we examined IL10 expression and found it changed significantly (Figure 4f; Page 18, Lines 10–14; Page 37, Lines 15–16). Since this is the first study to report the protective effect of VA on postmenopausal BMD, it is not unreasonable that there are still additional questions and research opportunities about the underlying mechanisms. We hope to provide this information for potential mechanism exploration in future studies.

Reviewer #2 (Remarks to the Author):

Xu Lin and colleagues have investigated the possible role of the gastrointestinal microbiome composition in bone mineral density. They associate the abundance of a single species *B. vulgatus* with BMD in their cohort and replicate this finding in an independent dataset. Furthermore, using multiple-omics *in vitro* and *in vivo* experiments they demonstrate the possible causality of *B. vulgatus* on BMD. The research and work presented is extensive and has great value for the field of BMD research, as the microbiome presents a modifiable risk factor which could be a novel therapeutic target for BMD (osteoporosis).

The authors have performed a tremendous amount of work and follow-up experiments to demonstrate the role of *B. vulgatus* in BMD. However, the statistical evidence of association for *B. vulgatus* is very weak, casting serious doubt on the interpretation and

conclusions drawn by Xu Lin and colleagues. This is mainly due to the (wrong) choice

of statistical methodology.

Major comments:

1. Consistently throughout the manuscript no multiple testing correction (Bonferroni, False Discovery Rate) is applied on p -values. How are the authors certain that their results are not spurious correlations? (Especially for the single microbiome species analysis this should have been done as still ~62 individual species were examined).

a. In the KEGG modules and BMD analysis results were significant if $p < 0.01$, why is the P -value threshold lowered here, and not in other analysis? Why not multiple testing correction in this analysis?

Response

We thank the reviewer for this comment. During the data analysis process, we calculated false discovery rate (FDR) for multiple testing correction on most of the p -values to be confident that the findings are not likely spurious correlations (Table 2 and 3). We did not perform FDR for p -values from constrained linear regression analysis (Table S1 and S5) since all bacteria species were tested in the same model and no multiple testing is involved.

In the KEGG modules and BMD analysis results, we performed multiple testing correction (FDR); however, since the q -values are > 0.050 and we could not perform *in vivo* or *in vitro* experiments for validation, we used a more stringent threshold than other analysis for determining significance (p -value < 0.010) for selecting the candidates for functional assays in future studies.

2. The authors consistently use statistical methods not suited for the analysis of compositional data such as metagenomics data. These traditional (non-compositional) methods can appear to give satisfactory results. However, these results can be misleading and unpredictable, thus I strongly recommend the authors to perform their association analysis using compositional methods(1).

a. This includes the Bray-Curtis dissimilarity matrix: the use of Bray-Curtis with compositional data (microbiome data) is very problematic, especially if the data has not been normalized (on read count)(1). (no mention of normalization procedures were found in the manuscript or extended data, nor average read count/range of read counts or if samples with low read count were excluded). The authors should have used methods which take the compositional nature of the microbiome data into account: log-ratio normalization then Aitchison distance matrix followed by the association analysis(MirKAT/PerANOVA ect.)

b. Also for the single microbiome species analysis methods which take the compositional nature of the microbiome data into account should have been used examples: MaAsLin (takes correlation between species into account) or isometric log-ratio transformation (to remove potential collinearity in the data)(1,2,3).

c. The same comment is also valid for the use Spearman correlation Coefficient, there are compositional methods available(1).

Response

We agree with the reviewer that metagenomics data requires compositional methods.

All the methods we used in this study are suitable for the compositional nature of the microbiome data. On one hand, we used Microbiome Regression-based Kernel Association Test (MiRKAT)¹³ to calculate bio-diversity (optimal kernel, based on weighted UniFrac, unweighted UniFrac, and Bray-Curtis distance metrics) and evaluated their associations with BMD (Page 6, Lines 6–9; Page 25 Line 21–Page 26 Line 5). On the other hand, we transformed the relative abundances of the species by using the centered log ratio (CLR) transformation, then applied constrained linear regression analysis for compositional data (e.g., relative abundance of bacterial species)¹⁴ (Page 6, Lines 19–21; Page 27, Lines 3–6). Therefore, we respectfully could not agree that our results are misleading and unpredictable, especially given the extensive validation by functional assays done as rigorous follow-ups.

3. Could the authors describe which “common probiotic bacteria” they have used and pooled for the correlation analysis with *B. vulgatus* (in the text or in the supplementary methods)? Could they also present a table/data set which contains the aver. abundancies of these “common probiotica bacteria”?

Response

We thank the reviewer for this comment. In our updated manuscript, we focus on VA-producing microbes (*Oscillibacter valericigenes*¹⁵ and *Megasphaera elsdenii*¹⁶) instead of the “common probiotic bacteria” we analyzed before because these two species are directly associated with VA production. Then we performed correlation analysis

between these two species and *B. vulgatus*.

The whole data (including the relative abundance of each species) that support the findings of this study are available from the corresponding author upon request and approval of the team and respective institutions once our manuscript is accepted (Page 37, Lines 18–21).

4. For the MR analysis a GWAS was performed to discover SNPs associated with *B. vulgatus* and valeric acid, however currently I am not able to assess the results of this analysis as essential information regarding this analysis is not available or not (clearly) presented in the text:

- a. What the sample size was for this GWAS, (how many individuals were whole genome sequenced and used for GWAS)
- b. On which individuals was this GWAS performed, was this the same set as of which metagenomics and phenotype data was available? (if yes this might cause for bias issues in the MR analysis) or did you use the published data of LLDeep (ref 43 in manuscript?)
- c. Were the selected SNPs both associated with *B. vulgatus* and valeric acid or one of the two?
- d. Which SNPs and how many SNPs were used in the MR analysis?
- e. Where are the summary statistics of the performed GWAS studies?

Response

We thank the reviewer for pointing out these questions. Whole genome sequencing was

performed by BGI Genomics Co. Ltd (Shenzhen, China; <https://www.genomics.cn/>) with BGISEQ-500 platform in the same Chinese cohort (n = 517). We used PLINK 1.9 software with the default criteria as follows to identify *B. vulgatus* and VA-associated SNPs: the SNPs with missing rate < 0.1, minor allele frequencies > 0.01, and Hardy-Weinberg equilibrium *p*-value > 1.0×10^{-5} were included for GWAS analysis (Page 33, Lines 9–13).

Since SCFAs are produced exclusively by the GM¹⁷, we investigated the causality of *B. vulgatus* exposure on VA outcome with one-sample MR analysis, which can provide unbiased estimates of the causal effect without the need for a traditional randomized clinical trial by using SNPs as instrumental variables (IVs)¹⁸. For the implementation of MR, we first selected independent genetic variants ($r^2 \leq 0.01$) associated with *B. vulgatus* (*p*-values < 1×10^{-5}) as the IVs. We then obtained the corresponding effect estimates of these IV SNPs from the VA GWAS analysis. To ensure the SNPs used as IVs for *B. vulgatus* are not in linkage disequilibrium (LD) with one another, a vital assumption of MR, we calculated pairwise-LD between all of our selected SNPs in the 1000 Genomes European reference sample using PLINK 1.9. For all the pairs of SNPs determined to violate the independence assumption with $r^2 > 0.01$, we retained only the SNPs with the smaller *B. vulgatus* association *p*-values. All of these contents were mentioned in the updated manuscript (Page 33 Line 3–Page 34 Line 5). The SNPs used in the MR analysis are shown in the Table S3. The whole data that support the findings of this study (including the summary statistics of the performed GWAS studies) are available from the corresponding author upon request and approval of the team and

respective institutions once our manuscript is accepted (Page 37, Lines 18–21).

5. *B. vulgatus* was only nominal associated with L1-L4-BMD (lumbar spine BMD, ED table 3). However, L1-L4 BMD was not correlated to BMD variation in Shannon index analysis and Kernal analysis (ED table 2). In addition, in the replication *B. vulgatus* was only nominal significant replicated in HTOT BMD (left total hip BMD). There is no consistent association of the microbiome or *B. vulgatus* with a single BMD site. Could the authors explain this discrepancy?

Response

Please refer to our responses to the second and the third questions of the first reviewer.

6. The authors are commended for taking the effort to replicate their main findings in an independent cohort, however this cohort has a very small sample size of 59. Perhaps the authors could consider collaborating with large microbiome cohorts and consortia for more robust replication? (MiBioGen, LLDEEP, HeLIOS, Rotterdam Study, TwinsUK etc) whom all have microbiomics and phenotypic data (some also have metagenomics and metabolomics).

Response

We thank the reviewer for this suggestion. Although the sample size of the Caucasian cohort was small, it was an available most closely matched sample for the validation.

All of the subjects were post-menopausal women. Individuals who had pathological conditions that may influence BMD (e.g., a bilateral oophorectomy, chronic renal failure, liver failure, lung diseases, gastrointestinal diseases, and inherited bone diseases), or may influence GM (e.g., taking antibiotics, having gastroenteritis, major surgery involving hospitalization, and inter-continental travel in the past three months) were excluded. For each participant, BMD was measured by using a Hologic Discovery-A DXA machine (Hologic Inc., Bedford, MA, USA) and information on age, medical history, physical activity, alcohol consumption, diet habits, smoking history, and nutrition supplements was assessed by a questionnaire. The accuracy of BMD measurement as assessed by the CV% of L1-L4 BMD was 0.54%. The metagenomic shotgun sequencing on the collected faecal samples was performed by Alkek Center for Metagenomics and Microbiome Research, Baylor College of Medicine (Page 28 Line 19–Page 29 Line 12).

Considering the above comments, it is my opinion that the current conclusions are not supported by the results, specifically the association of *B. vulgatus* with BMD, resulting in a manuscript unsuited for publication. However, it is also my opinion that if the authors address the above comments and using compositional methods find an association with *B. vulgatus* their manuscript can be an exceptionally good one.

Minor Comments

- The term “gut microbiome” although commonly used is actually not the correct term. It only refers to the microbial composition in the “gut”, while the stool microbiome is

actually a representative of the microbiome composition of the entire oral-gastrointestinal microbiome. Thus your gut microbiome composition here actually is: the relative abundance of faecal bacteriome. Please consider this in your manuscript, by for example defining this in the introduction or in the methods.

- Please give sample sizes in analysis and figures examples: how many non-rare species were there (abundance >0.10%)
- Figure 1: a nice figure but very difficult to read (especially for those who are colour blind like myself: Tritanomaly). Please consider only showing the distribution of genus level or different methods of representing all found species.
- The results of the single species association are now presented in the text, please consider to present these results in a table to increase readability (if journal constraints allow this).
- In the discussion the authors list the strength of their study, but do not mention any possible weaknesses or pitfalls of their own study. I find it good practice to also acknowledge own weaknesses or future improvement points of a study if strengths are mentioned.
- Please refrain from using such sentences: (discussion) we pioneered an innovative and comprehensive multi-omics approach for discovery in a Chinese cohort. Although you have done extensive research none of the used methods were novel or not published before.
- The discussion is quite lengthy and contains many repetitions, please consider restructuring the discussion to increase readability and to remove the unnecessary

repetition.

Response

We thank the reviewer for pointing out these detailed problems, and we have revised the entire manuscript accordingly. For example, we reorganized our figures/tables, and we mentioned our limitations in the updated manuscript.

References

- 1 Gregory B. Gloor, Jean M. Macklaim, Vera Pawlowsky-Glahn and Juan J. Egozcue, Microbiome Datasets Are Compositional: And This Is Not Optional. *Front. Microbiol.*, 15 November 2017
- 2 MaAsLin: <https://huttenhower.sph.harvard.edu/maaslin2/>
- 3 Aitchison, J., 1986, *The statistical analysis of compositional data: Monographs on statistics and applied probability*: Chapman & Hall Ltd., London, 416p.

Reviewer #3 (Remarks to the Author):

This is a fairly straightforward study that is clearly written and follows a logical flow. The analysis shows good bioinformatic and statistical knowhow. There are some concerns, however, that should be addressed.

1. As far as I can tell, while the study population seems substantial at first (517), only ~37 (7%) had osteoporosis. This is a small and probably underpowered number for metagenomic studies.

Response

We thank the reviewer for this comment. While many previous studies for osteoporosis have compared low/high BMD groups, the case-control design can often reduce statistical power due to loss of information from choosing an artificial cut-off value for group selection¹⁹. To make full use of the whole samples, and thus to have higher statistical power (compared with case-control analysis), we explored the association among GM, SCFA, and continuous BMD values (Page 4, Lines 9–14). Therefore, we do not need to consider the proportion of subjects with normal, osteopenia and osteoporosis.

2. Related to this, the data should be stratified for osteopenia and osteoporosis.

Response

Please refer to our responses to the second and the third questions.

3. In figure 1, it would help if the proportion of subjects with normal, osteopenia and osteoporosis were in some way projected onto the figure or shown separately/individually.

Response

Please refer to our responses to the second and the third questions.

4. It is not clear how the variables such as age and BMI vary with BMD. This should be represented or clarified in the text.

Response

We thank the reviewer for this comment. We have added Figure S1 to show associations between lifestyle factors (e.g., age, BMI, alcohol consumption, smoking, calcium supplementation, exercise)/socioeconomic status (e.g., education and family annual income) and BMD.

5. Figure 2: this should preferably be dot plot and not simply bars.

Response

We thank the reviewer for this suggestion. We have used boxplots (with median, interquartile range, and all data points) instead of bar graphs.

6. The authors also need to discuss the size of the effect *B. vulgatus* has on the BMD in the observational study.

Response

We thank the reviewer for this suggestion. In particular, *B. vulgatus* was found to be significantly associated with L1-L4 BMD ($\beta = -0.027$, p -value = 0.032, Figure 1c). The effect size of *B. vulgatus* had a similar magnitude of impact to other common BMD-related covariates, such as BMI, YSM, and exercise (Table S1), which suggests that this species may play an important role in BMD (Page 6 Line 22–Page 7 Line 5).

Minor points:

GM is a non-conventional abbreviation. It is unnecessary, will slow down readers and irritate many of them. It should be spelled out fully.

Response

We thank the reviewer for pointing out this detail, and we will revise the entire manuscript accordingly.

As far as I can tell, most if not all of the relevant literature is cited. However, for one of the earlier studies, *Das et al*, it is mentioned (line 255) that a trend of an association between shannon diversity and BMD was shown. As I recall, the *P*-value in that report was above 0.1 and so cannot be called a trend.

Response

We thank the reviewer for pointing this out. In the Fig. 3D of *Das et al* research²⁰, there was a trend of higher Shannon index (α -diversity) in osteoporosis and osteopenia groups compared with controls. Since the *P*-value in that report was above 0.1 (non-significant), we called it a “trend”.

References

- 1 Oksanen, J. *et al*. Package "vegan": Community Ecology package. *Time International* **1997**, 15–17 (2012).
- 2 Yang, T.-L. *et al*. Genetic and environmental correlations of bone mineral density at different skeletal sites in females and males. *Calcif Tissue Int.* **78**, 212-217 (2006).
- 3 Niu, T. *et al*. Identification of IDUA and WNT16 Phosphorylation-Related Non-Synonymous Polymorphisms for Bone Mineral Density in Meta-Analyses of Genome-Wide Association Studies. *J Bone Miner Res.* **31**, 358-368 (2016).
- 4 Koller, D. L. *et al*. Meta-analysis of genome-wide studies identifies WNT16 and ESR1 SNPs

- associated with bone mineral density in premenopausal women. *J Bone Miner Res.* **28**, 547-558 (2013).
- 5 Zheng, H.-F. *et al.* WNT16 influences bone mineral density, cortical bone thickness, bone strength, and osteoporotic fracture risk. *PLoS Genet.* **8**, e1002745 (2012).
- 6 Miller, P. D. Bone mineral density--clinical use and application. *Endocrinol Metab Clin North Am* **32**, 159-179 (2003).
- 7 Sprent, P. *Sign Test.* (Springer Berlin Heidelberg, 2011).
- 8 Payne, S. *et al.* In vitro studies on colonization resistance of the human gut microbiota to *Candida albicans* and the effects of tetracycline and *Lactobacillus plantarum* LPK. *Curr Issues Intest Microbiol.* **4**, 1-8 (2003).
- 9 Li, X. *et al.* Circular RNA CDR1as regulates osteoblastic differentiation of periodontal ligament stem cells via the miR-7/GDF5/SMAD and p38 MAPK signaling pathway. *Stem Cell Res Ther.* **9**, 232 (2018).
- 10 Gong, R. *et al.* Identification and Functional Characterization of Metabolites for Bone Mass in Peri- and Postmenopausal Chinese Women. *J Clin Endocrinol Metab.* **106**, e3159-e3177 (2021).
- 11 Park, J.-S., Lee, E.-J., Lee, J.-C., Kim, W.-K. & Kim, H.-S. Anti-inflammatory effects of short chain fatty acids in IFN- γ -stimulated RAW 264.7 murine macrophage cells: Involvement of NF- κ B and ERK signaling pathways. *Int Immunopharmacol.* **7**, 70-77 (2007).
- 12 Heyen, J. R. R., Ye, S. M., Finck, B. N. & Johnson, R. W. Interleukin (IL)-10 inhibits IL-6 production in microglia by preventing activation of NF- κ B. *Brain Res Mol Brain Res.* **77**, 138-147 (2000).
- 13 Zhao, N. *et al.* Testing in Microbiome-Profiling Studies with MiRKAT, the Microbiome Regression-Based Kernel Association Test. *Am J Hum Genet.* **96**, 797-807 (2015).
- 14 Lu, J., Shi, P. & Li, H. Generalized linear models with linear constraints for microbiome compositional data. *Biometrics* **75**, 235-244 (2019).
- 15 Yoshikawa, S. *et al.* Valerate production by *Megasphaera elsdenii* isolated from pig feces. *J Biosci Bioeng.* **125**, 519-524 (2018).
- 16 Iino, T., Mori, K., Tanaka, K., Suzuki, K.-I. & Harayama, S. *Oscillibacter valericigenes* gen. nov., sp. nov., a valerate-producing anaerobic bacterium isolated from the alimentary canal of a Japanese corbicula clam. *Int J Syst Evol Microbiol.* **57**, 1840-1845 (2007).
- 17 Lucas, S. *et al.* Short-chain fatty acids regulate systemic bone mass and protect from pathological bone loss. *Nat Commun.* **9**, 55 (2018).
- 18 Emdin, C. A., Khera, A. V. & Kathiresan, S. Mendelian Randomization. *JAMA* **318**, 1925-1926 (2017).
- 19 Yi, H. *et al.* Comparison of dimension reduction-based logistic regression models for case-control genome-wide association study: principal components analysis vs. partial least squares. *J Biomed Res.* **29**, 298-307 (2015).
- 20 Das, M. *et al.* Gut microbiota alterations associated with reduced bone mineral density in older adults. *Rheumatology (Oxford).* **58**, 2295-2304 (2019).

REVIEWER COMMENTS

Reviewer #1 (Remarks to the Author):

The authors have provided a rebuttal to my critique but no new analysis or experiments. My concerns still stand. I remained unconvinced about the validity of the conclusions of this study. The animal studies are weak. The authors have not added a sham operated group as requested in the original critique. Sham operated mice are needed to control for surgical stress. Quantitative histomorphometry is also needed. This analysis can be performed without tetracycline labelling, which is needed only to calculate dynamic indices of bone formation.

Reviewer #2 (Remarks to the Author):

I want to compliment the authors on their well written and extensive response. It is now very clear which methods the authors have used for the microbiome analysis and that they were quite rigorous in their statistical design.

All of my questions and comments were thoroughly answered, except for a few minor comments on the Genetic/MR analysis.

A) whole Genome Sequencing: some of the descriptives seem to be missing: sample size, the read depth and coverage of the whole genome sequencing samples.

B) I am missing some descriptives of the performed GWAS: sample size, nr of variants included after QC, and the QQplot (to see if the GWAS was not too underpowered or had inflation).

C) For the MR analysis several of the assumptions are very nicely tested for (pleiotropy), however I cannot find the test results for the testing of the strength of the MR instrument (F-statistic). As the GWAS sample size was very low (517) and none of the variants reached genome-wide significance it would be best to test if there is no "Weak instrument bias."

There is a handy checklist available for best practices for MR analysis reporting, which may be useful. <https://www.strobe-mr.org/>

[redacted]

We thank you for giving us an opportunity to further clarify and improve our manuscript. We also appreciate the reviewers for their valuable comments and helpful suggestions! Our responses to each reviewer's comments are summarized in the following, and the corresponding changes in the manuscript are highlighted in yellow.

Reviewer #1 (Remarks to the Author):

The authors have provided a rebuttal to my critique but no new analysis or experiments. My concerns still stand. I remained unconvinced about the validity of the conclusions of this study. The animal studies are weak. The authors have not added a sham operated group as requested in the original critique. Sham operated mice are needed to control for surgical stress. Quantitative histomorphometry is also needed. This analysis can be performed without tetracycline labelling, which is needed only to calculate dynamic indices of bone formation.

Response:

We thank the reviewer for the helpful suggestions! We have re-performed our mice *in vivo* experiments by adding the sham operated groups of mice as controls. Additionally, we increased the sample size (from n = 9/group to n = 12/group). After obtaining bone samples, we performed quantitative histomorphometry analysis

(including Von Kossa staining, tartrate-resistant acid phosphatase [TRAP] staining, hematoxylin-eosin [HE] staining, immunohistochemistry staining, and micro-computed tomography analysis) for quantitative indices of bone. All the updated results are shown in the revised Figures 2 and 3 below.

We observed poorer bone micro-structure in the ovariectomized (OVX)-normal saline (NS)/normal drinking water (ND)-mice, versus the sham-operated-NS/ND-mice (including lower Tb.N, Tb.Th and BV/TV but higher Tb.Sp, p -values < 0.001 , q -values < 0.001 , Figure 2f-2i and Figure 3f-3i), indicating bone loss in the OVX-mouse model and thus successful development of the OVX-mouse model. Similar results can be identified in the comparisons of the OVX-*Bacteroides vulgatus* (*B. vulgatus*)/VA-mice vs Sham-*B. vulgatus*/valeric acid (VA)-mice (results not shown but available if requested or needed).

Additionally, OVX mice gavaged with *B. vulgatus* demonstrated increased bone resorption and poorer bone micro-structure (compared with OVX-mice gavaged with normal saline), while feeding with valeric acid demonstrated reduced bone resorption and enhanced bone micro-structure (compared with OVX-mice fed with normal drinking water). The same/similar phenotype differences cannot be observed in the sham groups of mice, which indicates that the *B. vulgatus* has distinct effects in eugonadic mice and estrogen deficient mice.

Since menopause is a key period of change for women's health (increased risk of cardiovascular and metabolic diseases)¹ and *B. vulgatus* is an opportunistic pathogen²,

we speculate that the *B. vulgatus* produces a harmful effect on bone mineral density (BMD) mainly under the estrogen deficient status. This is consistent with reports that *B. vulgatus* is elevated with polycystic ovary syndrome-associated ovarian dysfunction³. We have added the related methods, results and discussion in the revised manuscript (Page 11, Line 1 to Page 13, Line 10; Page 16, Lines 14-21; Page 31, Line 3 to Page 34, Line 19; Page 46, Line 19 to Page 48, Line 12). Thank you!

Figure 2

Figure 2. *Bacteroides vulgatus* regulates bone-associated phenotypes of mice *in vivo*

vivo.

Changes in various bone-associated phenotypes in mice after gavage with *B. vulgatus* or NS for 8 weeks after OVX, compared with sham (non-OVX) and blank (non-OVX and no oral gavage) female mice (n = 12): (a)-(e) representative microCT/HE staining/Von Kossa staining/IHC-OC staining/TRAP staining images of the 5th lumbar vertebral body. Red/green arrows point to osteocalcin/osteoclasts positive cells, respectively. (f)-(i) quantitative indices of trabecular bone volume and structure, including Tb.N, Tb.Th, Tb.Sp, and BV/TV (n = 12). (j)-(l) quantitative data for percentages of Von Kossa⁺ area, osteocalcin positive cells, and osteoclasts (n = 6). (m)-(o) relative abundance of *B. vulgatus* and valeric acid-producing species (*Megasphaera elsdenii* and *Oscillibacter valericigenes*) (n = 12). (p)-(r) serum levels of valeric acid, PINP concentrations, and CTX-I concentrations (n = 12).

ns indicates non-significant, * indicates q -value < 0.05, ** indicates q -value < 0.01, *** indicates q -value < 0.001.

Figure 3. Valeric acid influences bone-associated phenotypes of mice *in vivo*.

Changes in various bone-associated phenotypes in mice after free-drinking with/without valeric acid for 8 weeks after OVX, compared with sham (non-OVX) and blank (non-OVX and no oral gavage) female mice ($n = 12$): (a)-(e) representative microCT/HE staining/Von Kossa staining/IHC-OC staining/TRAP staining images of the 5th lumbar vertebral body. Red/green arrows point to osteocalcin/osteoclast positive cells, respectively. (f)-(i) quantitative indices of trabecular bone volume and structure, including $Tb.N$, $Tb.Th$, $Tb.Sp$, and BV/TV ($n = 12$). (j)-(l) quantitative data for percentages of Von Kossa⁺ area, osteocalcin positive cells, and osteoclasts ($n = 6$). (m)-(o) serum levels of valeric acid, PINP concentrations, and CTX-I concentrations

(n = 12).

ns indicates non-significant, * indicates q -value < 0.05, ** indicates q -value < 0.01, *** indicates q -value < 0.001.

Abbreviations:

microCT - micro-computed tomography, B.V. - *Bacteroides vulgatus*, BV/TV - bone volume/tissue volume, CTX-I - C-telopeptide of type I collagen, HE - hematoxylin-eosin, IHC-OC - immunohistochemistry-osteocalcin, ND - normal drinking water without valeric acid treatment, NS - normal saline, OVX - ovariectomized, PINP - procollagen I N-terminal propeptide. Tb.N - trabecular number, Tb.Sp - trabecular separation, Tb.Th - trabecular thickness, TRAP - tartrate-resistant acid phosphatase, VA - valeric acid.

Reviewer #2 (Remarks to the Author):

I want to compliment the authors on their well written and extensive response. It is now very clear which methods the authors have used for the microbiome analysis and that they were quite rigorous in their statistical design.

All of my questions and comments were thoroughly answered, except for a few minor comments on the Genetic/MR analysis.

A) whole Genome Sequencing: some of the descriptives seem to be missing: sample size, the readdepth and coverage of the whole genome sequencing samples.

B) I am missing some descriptives of the performed GWAS: sample size, nr of variants included after QC, and the QQplot (to see if the GWAS was not too underpowered or had inflation).

C) For the MR analysis several of the assumptions are very nicely tested for (pleiotropy), however I cannot find the test results for the testing of the strength of the MR instrument (F-statistic). As the GWAS sample size was very low (517) and none of the variants reached genome-wide significance it would be best to test if there is no "Weak instrument bias".

There is a handy checklist available for best practices for MR analysis reporting, which may be useful. <https://www.strobe-mr.org/>

Response:

We thank the reviewer for your very helpful and constructive input.

A) The whole genome sequencing was performed by BGI Genomics Co. Ltd (Shenzhen, China; <https://www.genomics.cn/>) with BGISEQ-500 platform for the Chinese cohort (n= 500). After removing duplicate reads, the mean sequencing depth on the whole genome excluding gap regions was 17.69-fold. On average, per sequenced individual, 98.48% of the whole genome excluding gap regions were covered by at least 1X coverage, 96.85% had at least 4X coverage and 89.52% had at least 10X coverage. We have added this information to the revised manuscript (Page 8, Line 18 - Page 9, Line 2; Page 35, Lines 2-3). Thank you!

B) GWAS was performed for the same Chinese subjects mentioned above. There are 9,120,095 single nucleotide polymorphisms (SNPs) included after quality control. Based on the QQ plot of association statistics, we observed that there were likely minimal effects of confounding due to population structure (genomic inflation factor *Bacteroides vulgatus* = 1.016, genomic inflation factor_valeric acid = 1.028). We have added this information to the revised manuscript (Page 9, Lines 2 - 6; Page 35, Lines 10-12). The QQ plots were added in the supplementary information (Figure S2). Thank you.

C) We have calculated F-statistics to test the strength of the Mendelian randomization (MR) instrument⁴. The mean value of the F-statistics is 30.837 (F-statistic > 10),

indicating that there is no weak instrument bias. Detailed information was shown in the revised Table S3. We have added these contents in the revised manuscript (Page 9, Lines 12-13). Thank you.

References

- 1 Nappi, R. E., Chedraui, P., Lambrinoudaki, I. & Simoncini, T. Menopause: a cardiometabolic transition. *Lancet Diabetes Endocrinol.* **10**, 442-456 (2022).
- 2 Jeon, Y.-S., Chun, J. & Kim, B.-S. Identification of household bacterial community and analysis of species shared with human microbiome. *Curr Microbiol.* **67**, 557-563 (2013).
- 3 Qi, X. *et al.* Gut microbiota-bile acid-interleukin-22 axis orchestrates polycystic ovary syndrome. *Nat Med.* **25**, 1225-1233 (2019).
- 4 Burgess, S., Dudbridge, F. & Thompson, S. G. Combining information on multiple instrumental variables in Mendelian randomization: comparison of allele score and summarized data methods. *Stat Med.* **35**, 1880-1906 (2015).

REVIEWERS' COMMENTS

Reviewer #1 (Remarks to the Author):

I am satisfied with the reply and the new data except for a minor point. Von Koss area is not a standard histomorphometric index. Von Kossa staining is used to calculate bone area and BV/TV. Please use standard nomenclature (Dempster DW, Compston JE, Drezner MK, et al. Standardized nomenclature, symbols, and units for bone histomorphometry: a 2012 update of the report of the ASBMR Histomorphometry Nomenclature Committee. J Bone Miner Res. Jan 2013;28(1):2-17) and change the label of the y-axis in the applicable panels.

Reviewer #2 (Remarks to the Author):

No further comments

[redacted],

We thank you so much for giving us another opportunity to further clarify and improve our manuscript! We also appreciate the reviewers for their valuable comments and helpful suggestions! Our responses to each reviewer's comments are summarized in the following, and the corresponding changes in the manuscript are highlighted in yellow.

Reviewer #1 (Remarks to the Author):

I am satisfied with the reply and the new data except for a minor point. Von Koss area is not a standard histomorphometric index. Von Kossa staining is used to calculate bone area and BV/TV. Please use standard nomenclature (Dempster DW, Compston JE, Drezner MK, et al. Standardized nomenclature, symbols, and units for bone histomorphometry: a 2012 update of the report of the ASBMR Histomorphometry Nomenclature Committee. J Bone Miner Res. Jan 2013;28(1):2-17) and change the label of the y-axis in the applicable panels.

Response:

We thank the reviewer for the helpful suggestions! We learned from the recommended paper that mineralized volumes of bone could be quantified and labeled as percentages of mineralized area (Md. Ar) in Von Kossa staining. Therefore, we corrected the label "Von Kossa⁺ area (%)" to a standardized nomenclature "Md. Ar (%)" in the Fig.2j and Fig. 3j, as well as in the manuscript (Lines 11-12 in Page 11, Lines 21-22 in Page 12, Lines 6-8 in Page 55, and Lines 12-13 in Page 56). Updated

figures are shown below. Thank you!

Figure 2

(a)-(e) representative microCT/HE staining/Von Kossa staining/IHC-OC staining/TRAP staining images of the 5th lumbar vertebral body. Arrows point to osteocalcin positive cells (d) and osteoclasts positive cells (e). (f)-(i) quantitative indices of trabecular bone volume and structure, including Tb.N, Tb.Th, Tb.Sp, and BV/TV (n_mice = 12/group). (j) mineralized volumes of mice bone are quantified by percentages of mineralized area in Von Kossa staining (n_mice = 6/group). (k) and (l)

quantitative data of osteocalcin positive cells and osteoclasts, respectively (n_mice = 6/group). (m)-(o) relative abundance of *B. vulgatus* and valeric acid-producing species (*Megasphaera elsdenii* and *Oscillibacter valericigenes*) (n_mice = 12/group). (p)-(r) serum levels of valeric acid, PINP concentrations, and CTX-I concentrations (n_mice = 12/group).

(a)-(e) representative microCT/HE staining/Von Kossa staining/IHC-OC staining/TRAP staining images of the 5th lumbar vertebral body. Arrows point to osteocalcin positive cells (d) and osteoclasts positive cells (e). (f)-(i) quantitative indices of trabecular bone volume and structure, including Tb.N, Tb.Th, Tb.Sp, and BV/TV (n_mice = 12/group). (j) mineralized volumes of mice bone are quantified by

percentages of mineralized area in Von Kossa staining (n_mice = 6/group). (k) and (l) quantitative data of osteocalcin positive cells and osteoclasts, respectively (n_mice = 6/group). (m)-(o) serum levels of valeric acid, PINP concentrations, and CTX-I concentrations (n_mice = 12/group).

Reviewer #2 (Remarks to the Author):

No further comments.

Response:

Thank you!